# Greedy Optimization Provably Wins the Lottery: Logarithmic Number of Winning Tickets is Enough

**Mao Ye** [*]
UT Austin
my21@cs.utexas.edu

**Lemeng Wu** [*]
UT Austin
lmwu@cs.utexas.edu

**Qiang Liu**
UT Austin
lqiang@cs.utexas.edu

## Abstract

Despite the great success of deep learning, recent works show that large deep neural networks are often highly redundant and can be significantly reduced in size. However, the theoretical question of how much we can prune a neural network given a specified tolerance of accuracy drop is still open. This paper provides one answer to this question by proposing a greedy optimization based pruning method. The proposed method has the guarantee that the discrepancy between the pruned network and the original network decays with exponentially fast rate w.r.t. the size of the pruned network, under weak assumptions that apply for most practical settings. Empirically, our method improves prior arts on pruning various network architectures including ResNet, MobilenetV2/V3 on ImageNet.

## 1 Introduction

Large-scale deep neural networks have achieved remarkable success on complex cognitive tasks, including image classification, e.g., He et al. (2016), speech recognition, e.g., Amodei et al. (2016) and machine translation, e.g.,Wu et al. (2016). However, a drawback of the modern large-scale DNNs is their low inference speed and high energy cost, which makes it less appealing to deploy those models on edge devices such as mobile phones and Internet of Things (Cai et al., 2019).

It has been shown that network pruning (Han et al., 2015) is an effective technique to reduce the size of the DNNs without a significant drop of accuracy. However, most existing works on network pruning are based on heuristics, leaving the theoretical questions largely open on what kind of network can be effectively pruned, how much we can prune a DNN given a specified tolerance of accuracy drop and how to achieve it with a practical and computationally efficient procedure.

Recently, a line of works on network pruning with theoretical guarantees have emerged, including sensitivity-based methods (Baykal et al., 2019b; Liebenwein et al., 2020), coreset-based methods (Baykal et al., 2019a; Mussay et al., 2020), greedy forward selection (Ye et al., 2020). Both the sensitivity-based and coreset-based methods prune the network by sampling and bound the error caused pruning via concentration inequalities. They show that the error introduced by pruning decays with an $\mathcal{O}(n^{-1})$ rate w.r.t. the size $n$ of pruned network. This is comparable to the asymptotic error obtained by directly training a neural network of size $n$ with gradient descent descent, which is also $\mathcal{O}(n^{-1})$ following the mean field analysis of Mei et al. (2018); Araújo et al. (2019); Sirignano & Spiliopoulos (2019). More recently, Ye et al. (2020) proposed the first pruning method that achieves a faster $\mathcal{O}(n^{-2})$ error rate and is hence provably better than direct training with gradient descent. See Table 1 for a summary on those works.

However, the analysis of Ye et al. (2020) only applies to two-layer networks and requires the original network to be sufficiently over-parameterized. In this paper, we proposed a new greedy optimization based pruning method, which learns sub-networks of size $n$ with a significantly smaller $\mathcal{O}(\exp(-cn))$

---

[*]Equal Contribution

| | Rate | No Over-param | Deep Net |
|---|---|---|---|
| Baykal et al. (2019b); Liebenwein et al. (2020) | $\mathcal{O}(n^{-1})$ | ✓ | ✓ |
| Baykal et al. (2019a); Mussay et al. (2020) | $\mathcal{O}(n^{-1})$ | ✓ | ✓ |
| Ye et al. (2020) | $\mathcal{O}(n^{-2})$ | ✗ | ✗ |
| This paper | $\mathcal{O}(\exp(-cn))$ | ✓ | ✓ |

Table 1: Overview on theoretical guaranteed pruning methods. Rate above gives how the error due to pruning decays as the size of the pruned network ($n$) increases. Column 'No Over-param' denotes whether the method applies to an original network that is not over-parameterized in order to obtained the rate. Column 'Deep net' denotes whether the analysis applies to deep networks.

error rate, improving the rate from polynomial to exponential. In addition, our theoretical rate only requires weak assumptions that hold for most networks in practice, without requiring the the original networks to be overparameterized as Ye et al. (2020). Different from the Lottery Ticket Hypothesis (Frankle & Carbin, 2018), which selects the winning tickets that give good performance when trained in isolation from initialization, our approach finds the tickets (that already won) from a fully converged network.

Practically, our algorithm is simple and easy to implement. In addition, we introduce practical speedup techniques to further improve the time efficiency. Empirically, our method improves the prior arts on network pruning under various network structures including ResNet-34 (He et al., 2016), MobileNetV2 (Sandler et al., 2018) and MobileNetV3 (Howard et al., 2019) on ImageNet (Deng et al., 2009) as well as DGCNN (Wang et al., 2019) on ModelNet40 (Wu et al., 2015) on point cloud classification.

**Notation**  We use notation $[N] := 1, ..., N$ for the set of the first $N$ positive integers. All the vector norms $\|\cdot\|$ are assumed to be $\ell_2$ norm. We denote the vector $\ell_0$ norm by $\|\cdot\|_0$. $\|\cdot\|_{\text{Lip}}$ denotes the Lipschitz norm for functions. $\mathbb{I}\{\cdot\}$ indicates the indicator function.

## 2  Background and Method

**Problem Setup**  Given a pre-trained deep neural network with $L$ layers: $F(\boldsymbol{x}) = F_L \circ F_{L-1} \circ \cdots \circ F_2 \circ F_1(\boldsymbol{x})$, where the $\ell$-th layer $F_\ell$ consisting of $N$ neurons forms a mapping of form

$$F_\ell(\boldsymbol{z}) = \frac{1}{N} \sum_{i=1}^{N} \sigma(\boldsymbol{\theta}_i^\ell, \boldsymbol{z}),$$

with $\boldsymbol{z}$ as a proper input of the $\ell$-th layer, which is the output of the previous $\ell - 1$ layers. Here $\sigma(\boldsymbol{\theta}, \cdot)$ is a general nonlinear map parameterized by $\boldsymbol{\theta}$ that represents a neuron or other module in the network. For example, in a fully connected layer, we have $\sigma(\boldsymbol{\theta}, \boldsymbol{z}) = w_1 \sigma_+(\boldsymbol{w}_2^\top \boldsymbol{z})$ with $\boldsymbol{\theta} = [w_1, \boldsymbol{w}_2]$ and $\sigma_+$ an activation function such as ReLU or Tanh. In a convolution layer, we have $\sigma(\boldsymbol{\theta}, \boldsymbol{z}) = w_1 \sigma_+(\boldsymbol{w}_2 * \boldsymbol{z})$, where $*$ denotes the convolution operator. In this paper we may call $\sigma(\boldsymbol{\theta}_i^\ell, \cdot)$ the neuron $i$ or the $i$-th neuron for simplicity. Without loss of generality, we assume each layer in the given deep network has the same number of neurons using the same activation function.

The goal of network pruning is to construct a thinner network by replacing each layer with a subset of $n < N$ neurons. For simplicity of presentation, we focus on pruning a single layer $F_\ell$ for now and we discuss how to apply our algorithm in a layer-wise fashion to prune the whole network in section 2.4.

To prune the $\ell$-th layer, the goal is to replace $F_\ell$ with a thinner layer $f_{\ell, \boldsymbol{A}}$ with $n < N$ neurons:

$$f_{\ell, \boldsymbol{A}}(\boldsymbol{z}) = \sum_{i=1}^{N} a_i \sigma(\boldsymbol{\theta}_i^\ell, \boldsymbol{z}), \quad \boldsymbol{A} = [a_1, ..., a_N] \in \Omega_N, \|\boldsymbol{A}\|_0 \leq n,$$

where $\Omega_{[N]}$ is the probability simplex on the $N$ neurons, that is,

$$\Omega_N = \left\{ \boldsymbol{v} : \boldsymbol{v} = [v_1, ..., v_N] \in \mathbb{R}^N, \quad v_i \geq 0, \quad \forall i \in [N] \quad \text{and} \quad \sum_{i=1}^{N} v_i = 1 \right\}.$$

By enforcing that $\boldsymbol{A} \in \Omega_N$, we prune the layer by finding the best convex combination of a subset of neurons. The constraint that $\sum_i a_i = 1$ ensures that the overall magnitude of the output of the layer after pruning matches that of the original network even when a lot neurons are moved. We denote the network with the $\ell$-th layer replaced by $f_{\ell, A}$ as $f_A$, i.e.,

$$f_A = F_L \circ ... \circ F_{\ell+1} \circ f_{\ell, A} \circ F_{\ell-1} \circ ... \circ F_1.$$

Given an observed dataset $\mathcal{D}_m := (\boldsymbol{x}^{(i)}, y^{(i)})_{i=1}^m$ with $m$ data points. We want to choose $\boldsymbol{A}$ such that the pruned network $f_{\boldsymbol{A}}$ is close to the original $F$ as much as possible, measured by the regression discrepancy loss,

$$\mathbb{D}[f_{\boldsymbol{A}}, \ F] = \mathbb{E}_{\boldsymbol{x} \sim \mathcal{D}_m} \left[ (f_A(\boldsymbol{x}) - F(\boldsymbol{x}))^2 \right].$$

Our algorithm and theoretical analysis can be extended to other discrepancy losses such as the cross-entropy. The problem of pruning the $\ell$-th layer can be formulated by the following constraint problem

$$\min_{\boldsymbol{A}} \ \mathbb{D}[f_{\boldsymbol{A}}, F], \ \text{s.t.} \quad \boldsymbol{A} \in \Omega_N, \quad \|\boldsymbol{A}\|_0 \leq n. \tag{1}$$

This yields a challenging sparse optimization problem, which we address using greedy optimization, yielding algorithms that are both theoretically guaranteed and practically efficient.

## 2.1 Pruning with Greedy Local Imitation

We first introduce a simply greedy algorithm via local imitation for searching a good solution of problem (1). The pruned network can be viewed as

$$H \circ f_{\ell, \boldsymbol{A}}(\boldsymbol{z}) = H \circ \left( \sum_{i=1}^N a_i \sigma(\boldsymbol{\theta}_i^\ell, \boldsymbol{z}) \right).$$

Here $\boldsymbol{z}$ is the output of the $\ell - 1$-th layers and $H = F_L \circ ... \circ F_{\ell+1}$ is the mapping of the later layers. Denote $\boldsymbol{z}^{(i)} = F_{\ell-1} \circ ... \circ F_1(\boldsymbol{x}^{(i)})$. The set $\mathcal{D}_m^\ell := (\boldsymbol{z}^{(i)})_{i=1}^m$ denotes the distribution of training data pushed through the first $\ell - 1$ layers. Suppose $H$ is Lipschitz continuous, which typically holds for neural networks, we are about to upper bound $\mathbb{D}[f_{\boldsymbol{A}}, F]$ by $\mathbb{D}[f_{\boldsymbol{A}}, F] \leq \|H\|_{\text{Lip}}^2 \bar{\mathbb{D}}[f_{\ell, \boldsymbol{A}}, F_\ell]$, where $\bar{\mathbb{D}}$ is the local discrepancy loss measuring the discrepancy on the output of the $\ell$-th layer between the pruned and original network

$$\bar{\mathbb{D}}[f_{\ell, \boldsymbol{A}}, F_\ell] = \mathbb{E}_{\boldsymbol{z} \sim \mathcal{D}_m^\ell} \|f_{\ell, \boldsymbol{A}}(\boldsymbol{z}) - F_\ell(\boldsymbol{z}))\|^2.$$

In local imitation, we construct $f_{\ell, A}$ such that its output well imitates the output of $F_\ell$, i.e.,

$$\min_{\boldsymbol{A}} \ \bar{\mathbb{D}}[f_{\ell, \boldsymbol{A}}, F_\ell], \ \text{s.t.} \quad \boldsymbol{A} \in \Omega_N, \quad \|\boldsymbol{A}\|_0 \leq n. \tag{2}$$

Importantly, different from the original loss $\mathbb{D}[\cdot]$ in (1), the layer-wise local discrepancy loss $\bar{\mathbb{D}}[\cdot]$ is convex w.r.t. $\boldsymbol{A}$ and enjoys good geometric property for enabling fast exponential error rate via greedy optimization, as we show in sequel. On the other hand, as the final discrepancy $\mathbb{D}$ is controlled by the local discrepancy $\bar{\mathbb{D}}$, i.e., minimizing $\bar{\mathbb{D}}$ effectively minimizes $\mathbb{D}$.

The local imitation is a bi-directional greedy optimization for solving (2). It starts with an empty layer, and sequentially adds, removes or adjusts neurons that yield the largest decrease of the loss. Specifically, denote by $f_{\ell, \boldsymbol{A}(k)}$ the layer we obtained at the $k$-th iteration with $\boldsymbol{A}(k) = [a_1(k), ..., a_N(k)]$. We start with selecting the single best neuron that minimizes the loss:

$$f_{\ell, \boldsymbol{A}(0)} = \sigma(\boldsymbol{\theta}_{i_0^*}^\ell, \cdot), \quad \text{with} \quad i_0^* = \arg\min_{i \in [N]} \bar{\mathbb{D}}[\sigma(\boldsymbol{\theta}_i^\ell, \cdot), \ F_\ell(\cdot)] \tag{3}$$

where $i_0^*$ is the index of the selected neuron; correspondingly, we have $a_i(0) = \mathbb{I}\{i = i_0^*\}$.

At iteration $k$, we search for the best neuron $i_k^*$ and step size $\gamma_k^*$ that minimizes the loss most, i.e.,

$$[i_k^*, \gamma_k^*] = \arg\min_{i \in [N], \gamma \in U_i} \bar{\mathbb{D}}\left[ (1 - \gamma) f_{\ell, \boldsymbol{A}(k)} + \gamma \sigma(\boldsymbol{\theta}_i^\ell, \cdot), \ F_\ell \right]. \tag{4}$$

Then we update $f_{\ell,\boldsymbol{A}(k)}$ to $f_{\ell,\boldsymbol{A}(k+1)} = (1 - \gamma_k^*) f_{\ell,\boldsymbol{A}(k)} + \gamma_k^* \sigma(\boldsymbol{\theta}_i^\ell, \cdot)$. Here $U_i$ in (4) is the search interval of the step size $\gamma$. We set $U_i = [0, 1]$ if the $i$-th neuron has not been selected yet (i.e., $a_i(k) = 0$) and $U_i = [-a_i(k)/(1 - a_i(k)), 1]$ if the neuron has already been added (i.e., $a_i(k) > 0$). Therefore, this update can correspond to adding or removing a neuron, or simply adjusting the weight of existing neurons: the $i_k^*$-th neuron is added into the pruned network $f_{\ell,\boldsymbol{A}}$ if we have $a_{i_k^*}(k) = 0$, and it is removed from $f_{\ell,\boldsymbol{A}}$ if we have $\gamma_k^* = -a_{i_k^*}(k)/(1 - a_{i_k^*}(k))$; no new neuron is added or removed if otherwise.

We stop the iteration when a convergence criterion, i.e., $\bar{\mathbb{D}} \leq \epsilon$, is met.

**Solving Greedy Optimization in** (4)    A naive way to solve problem (4) is by enumerating each neuron and solving the corresponding inner minimization on $\gamma$. This is computational costly as it requires computing the forward pass in neural network many times. However, given $i$, the local discrepancy loss is a quadratic function w.r.t. $\gamma$. Combined with some special property of the local imitation algorithms, we are able to solve (4) with only computing the forward pass in network once. We refer readers to Appendix 5.1 for details

### 2.1.1   Greedy Local Imitation Decays Error Exponentially Fast

Now we proceed to give the convergence rate for the proposed local imitation algorithm. We introduce the following assumption.

**Assumption 1** *Assume that for any $i \in [N]$, $\boldsymbol{z}^{(j)} \in \mathcal{D}_m^\ell$, we have $\left\| \sigma(\boldsymbol{\theta}_i^\ell, \boldsymbol{z}^{(j)}) \right\| \leq c_1$ and $\|H\|_{Lip} \leq c_1$ for some $c_1 < \infty$.*

Assumption 1 holds when network parameters and data are bounded and the activation is Lipschitz continuous, which is very mild and holds for most network in practice. The following theorem characterizes the convergence of local imitation showing that the error caused by pruning decays exponentially fast when the number of neurons in the pruned model increases.

**Theorem 1 (Convergence Rate)** *Under assumption 1, at each step $k$ of the greedy optimization in (2), we obtain a layer with no more than $k$ neurons (i.e., $\|\boldsymbol{A}(k)\|_0 \leq k$), whose loss satisfies $\mathbb{D}[f_{\boldsymbol{A}(k)}, F] \leq \|H\|_{Lip}^2 \bar{\mathbb{D}}[f_{\ell,\boldsymbol{A}(k)}, F_\ell] = \mathcal{O}\left(\exp(-\lambda_\ell k)\right)$, where $\lambda_\ell > 0$ is a strictly positive constant. That is, the loss decays exponentially with the number of neurons in $f_{\ell,\boldsymbol{A}(k)}$.*

**Remark**    Minimizing $\gamma$ over $U_i$ can be viewed as line searching the optimal step size for adjusting neuron $i$. We may also consider to choose a proper fixed step size, e.g., $\gamma = 1/(k+1)$ instead of line searching, in which case the optimization in (2) is simplified into

$$\min_{i \in [N]} \bar{\mathbb{D}} \left[ \left( k f_{\ell,\boldsymbol{A}(k)} + \sigma(\boldsymbol{\theta}_i^\ell, \cdot) \right) / (k+1), \ F_\ell \right], \tag{5}$$

which can be shown to give an $\mathcal{O}(k^{-2})$ error at the $k$-th step under the same assumption as Theorem 1. See Appendix 5.2 for more details.

## 2.2   Pruning with Greedy Global Imitation

The local imitation method uses a surrogate local discrepancy loss which is convex w.r.t. $\boldsymbol{A}$ to prune the networks. Despite the good property of local imitation, the use of surrogate loss can be ineffective for some layers. For example, during the iteration of local imitation, the best neuron that minimizes the surrogate loss is not necessarily the best one that minimizes the actual discrepancy loss.

We propose a second pruning method, which directly minimizing the original discrepancy loss. This method follows the similar greedy fashion as the local imitation. We initialize the network by $f_{\boldsymbol{A}(0)} = H_2 \circ f_{\ell,\boldsymbol{A}(0)} \circ H_1 = H_2 \circ \left( \sum_{i=1}^N a_i(0) \sigma(\boldsymbol{\theta}_i^\ell, \cdot) \right) \circ H_1$, where

$$a_i(0) = \mathbb{I}\{i = i_0^*\}, \quad i_0^* = \arg\min_{i \in [N]} \mathbb{D}[H_2 \circ \sigma(\boldsymbol{\theta}_i^\ell, \cdot) \circ H_1, F], \tag{6}$$

and $H_1 = F_{\ell-1} \circ \cdots \circ F_1$ and $H_2 := F_L \circ \cdots \circ F_{\ell+1}$. Similarly, at each iteration, We adjust the network in a greedy way by solving the following problem:

$$\min_{i \in ([N]} \min_{\gamma \in U_i} \mathbb{D} \left[ H_2 \circ \left( (1 - \gamma) f_{\ell,\boldsymbol{A}(k)} + \gamma \sigma(\boldsymbol{\theta}_i^\ell, \cdot) \right) \circ H_1, F \right]. \tag{7}$$

However, solving problem (7) is computationally costly as the loss is non-convex w.r.t. $\boldsymbol{A}$ and thus solving the inner minimization on $\gamma$ requires exhaustive search. To reduce the computational cost, in iteration $k$, we instead consider the following problem

$$\min_{i \in [N]} \mathbb{D}\left[H_2 \circ \left((1-\gamma_k) f_{\ell, \boldsymbol{A}(k)} + \gamma_k \sigma(\boldsymbol{\theta}_i^\ell, \cdot)\right) \circ H_1, F\right], \ \gamma_k = (1+k)^{-1}. \tag{8}$$

Suppose that $i_k^*$ gives that solution of problem (8), we update the network by setting

$$a_i(k+1) = (1-\gamma_k) a_i(k) + \gamma_k \mathbb{I}\{i = i_k^*\}.$$

And we end the iteration when convergence criterion is met. Notice that different from local imitation, the algorithm adjusts $A$ based on the final output of the network instead of the 'local' output of the pruned layer and thus we name it global imitation.

Different from the local imitation, due to the nonlinear of $H_2$, besides Assumption 1, obtaining a convergence rate for the global imitation requires several additional assumptions characterizing the linearity of $H_2$ as well as the geometric property of the pruned layer.

**Theorem 2** *Under Assumption 1 and some additional assumptions, specified in Appendix 5.3, on the linearity of $H_2$ and initialization, we have $\mathbb{D}[f_{\boldsymbol{A}(k)}, F] = \mathcal{O}(k^{-2})$ and $\|\boldsymbol{A}(k)\|_0 \leq k$.*

### 2.2.1 Accelerating Global Imitation via Taylor Approximation

A native way to solve problem (8) is by enumerating all the neurons and calculating $\mathbb{D}\left[H \circ \left((1-\gamma_k) f_{\ell, \boldsymbol{A}(k)} + \gamma_k \sigma(\boldsymbol{\theta}_i^\ell, \cdot)\right), F\right]$, which has at least $\mathcal{O}(Nn)$ time complexity for pruning a layer with $N$ neurons to $n$ neurons. Here we propose a technique to reduce the computational cost via Taylor approximation. At iteration $k$, for any neuron $i \in [N]$, we have

$$\mathbb{D}\left[H \circ \left[(1-\gamma_k) f_{\ell, \boldsymbol{A}(k)} + \gamma_k \sigma(\boldsymbol{\theta}_i^\ell, \cdot)\right], F\right] = \frac{1}{k+1} gr_{\boldsymbol{A}(k), i} + \mathcal{O}\left((k+1)^{-2}\right),$$

where we define

$$gr_{\boldsymbol{A}(k), i} = \frac{\partial}{\partial \gamma} \mathbb{D}\left[H \circ \left[(1-\gamma) f_{\ell, A(k)} + \gamma \sigma(\boldsymbol{\theta}_i, \cdot)\right], F\right]\bigg|_{\gamma=0}.$$

Thus, when $k$ is large enough (which we find 25 is sufficient in practice), this approximation allows us to find the (near) optimal solution with small error of problem (8) by finding the neuron with the largest $gr_{\boldsymbol{A}, i}$. Simple algebra shows that

$$gr_{\boldsymbol{A}, i} = 2 \sum_{j=1}^n \left(\mathbb{I}\{j = i\} - a_j\right) r_{\boldsymbol{A}, i}, \quad \text{where}$$

$$r_{\boldsymbol{A}, i} := \mathbb{E}_{\boldsymbol{z} \sim \mathcal{D}_m^\ell} \left[\left(H \circ f_{\ell, \boldsymbol{A}}(\boldsymbol{z}) - H \circ F_\ell(\boldsymbol{z})\right) H'(f_{\ell, \boldsymbol{A}}(\boldsymbol{z})) \sigma(\boldsymbol{\theta}_i^\ell, \boldsymbol{z})\right].$$

Therefore, we can easily calculate $gr_{\boldsymbol{A}(k), i}$ for all $i \in [N]$ once we obtain $r_{\boldsymbol{A}(k), i}$ for all $i \in [N]$. In appendix, we show that $r_{\boldsymbol{A}(k), i}$ can be easily computed with automatic differentiation function in common deep learning libraries by introducing some ancillary parameters into the model. See Appendix 5.4 for details. If we choose to use this approximation when $k > \tilde{k}$ for some $\tilde{k} > 0$, we reduce the complexity from $\mathcal{O}(Nn)$ to $\mathcal{O}(n)$.

### 2.3 Pruning vs GD: Numerical Verification of the Rate

Our result implies that the subnetwork $f_{\boldsymbol{A}}$ obtained by pruning gives $\mathbb{D}[f_{\boldsymbol{A}}, F] = \mathcal{O}\left(\exp(-\lambda n)\right)$ where $n$ is the number of neurons remained in the pruned layer. In comparison, the mean field analysis (Araújo et al., 2019; Mei et al., 2018) suggests that directly train a network with same size as the pruned model gives $\mathcal{O}\left(n^{-1}\right)$ discrepancy loss. This suggests that pruning is provably better than training. We conduct a numerical experiment to verify the theoretical result. Given some simulated dataset, we firstly train a two hidden layer neural network with 100 neurons for each layer. And we prune the layer close to the input to different number of neurons using the local and global imitation. We also train the network with different number of neurons for the pruned layer and 100 neurons for the other one. Figure 1 plots the discrepancy loss and the number of neurons of the pruned layer. The empirical result matches our theoretical findings. We refer readers to Appendix 5.5 for more details.

**Algorithm 1** The Greedy Local/Global Imitation

1: **Input**: A pretrained network $F$ with $L$ layers. The targeted layer index $\ell$ for pruning. Method = $\in$ {local, global}
2: Initialize $f_{\ell,\boldsymbol{A}}$ using (3) for local imitation else (6) for global imitation.
3: **while** convergence criterion is not met **do**
4:    Randomly sample a mini-batch data $\hat{\mathcal{D}}$.
5:    Update $f_{\ell,\boldsymbol{A}}$ by solving (4) for local imitation or (8) for global imitation, using data $\hat{\mathcal{D}}$.
6: **end while**
7: **Return:** The pruned layer $f_{\ell,\boldsymbol{A}}$.

---

**Algorithm 2** Layer-wise Prune

1: **Input**: pretrained network $F$ with $L$ layers.
2: **for** $\ell = 1, 2, ..., L$ **do**
3:    Obtain the pruned layer $f_{\ell,\boldsymbol{A}^{\text{local}}}$ by local imitation on $F$ with target layer $\ell$.
4:    Obtain the pruned layer $f_{\ell,\boldsymbol{A}^{\text{global}}}$ by global imitation on $F$ with target layer $\ell$.
5:    Replace the $\ell$-th layer $F_\ell$ of $F$ with $f_{\ell,\boldsymbol{A}^{\text{local}}}$ if local imitation is better, else $f_{\ell,\boldsymbol{A}^{\text{global}}}$.
6: **end for**
7: **Return:** The pruned network $F$.

## 2.4 Practical Algorithm: Pruning All Layers

In section 2.1 and 2.2 we introduce how to use the greedy local/global imitation to prune a certain layer in a network. In order to prune the whole network, we apply the greedy optimization scheme in a layer-wise fashion. Starting from the full network $F$, we apply the pruning method to prune the first layer (the one that is closest to the input) $F_1$ to $f_1$, which returns a pruned network $F^{\text{prune},1} = F_L \circ F_{L-1} \circ \cdots \circ F_2 \circ f_1$. We then apply the pruning method to prune the second layer $F_2$ in $F^{\text{prune},1}$ and continue until all the layers are pruned. By applying the pruning algorithm in this manner we prune the whole network.

The local imitation and global imitation perform differently when pruning different layers. To combine their advantages, when pruning each layer, both methods are applied individually with same convergence criterion and the one gives better performance is picked up. In this paper, we stop pruning when the discrepancy loss $\mathbb{D}$ of the pruned model is smaller than a user specified threshold. The method prunes more neurons at convergence is selected. If both methods prune the same number of neurons, then the one with smaller discrepancy loss is chosen. Algorithm 1 summarizes the procedure of local and global imitation and Algorithm 2 gives the layer-wise scheme on pruning the whole deep network.

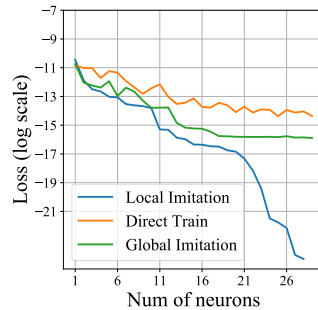

Figure 1: Discrepancy loss of the pruned model and train-from-scratch network with different sizes. The loss is in logarithm scale.

The exponential decay rate can be obtained by iteratively applying our theory on each layer. Suppose the pruned model $F^{\text{pruned}}$ has $n$ neurons at each layer, we have $\mathbb{D}[F^{\text{pruned}}, F] = \mathcal{O}(\exp(-\lambda n))$. We refer reader to Appendix 5.6 for details. Also notice that the exponential decay rate also holds for Algorithm 1 as it chooses the method with smaller loss.

# 3 Experiment

## 3.1 Comparing the Local and Global Imitation

Our first experiment aims to analyze the performance of local and global imitation for pruning deep neural network for image classification. We first apply both methods to a pretrained VGG-11 on CIFAR-10 dataset. We prune all the 8 convolution layers individual (when pruning one layer, the

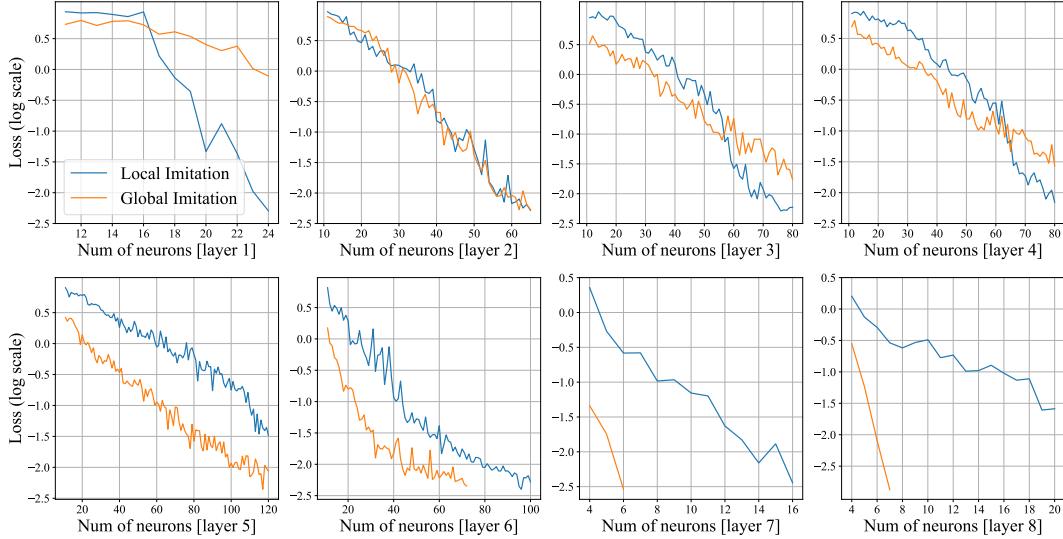

Figure 2: Pruning convolution layers on VGG11 using local and global imitation. From left to right and from top to bottom corresponds to the first (the one close to input) to the last convolution layers.

other layers remain unpruned) using both local and global imitation in order to compare these two methods side by side.

**Settings** The full network is trained with SGD optimizer with momentum 0.9. We use 128 batch size with initial learning rate 0.1 and train the model for 160 epochs. We decay the learning rate by 0.1 at the 80-th and the 120-th epochs. During pruning we use 128 batch size. We do not apply the Taylor approximation tricks to global imitation for this experiment. we use cross entropy between the pruned model and original model as discrepancy loss.

**Result** Figure 2 summarizes the result. Overall, we find the local and global imitation performs differently on different layers. The local imitation tends to decreases the loss faster on layer that is more close to input and with less neurons. While global imitation tends to performs much better than local imitation on layer that are close to output.

**Combining Local and Global Imitation Outperforms Both** In practice, we find purely pruning with local imitation tends gives worse result than pruning only with global imitation. However, combining the local imitation with global imitation performs better than pruning only with global imitation. To show this, we apply local and global imitation with the same setting as in section 2.4 on pruning ResNet34 and MobileNetV2 on ImageNet. For comparison, pruning with only global imitation is also applied. The local+global setting achieves 73.5 top1 accuracy on the pruned ResNet34 with 2.2G FLOPs and 72.2 top1 accuracy on the pruned MobileNetV2 with 245M FLOPs. While pruning with only global imitation only achieve 73.2 top1 accuracy on ResNet34 and 72.1 top1 accuracy on MobileNetV2 with same FLOPs. The experimental settings are in Section 3.2.

## 3.2 Imagenet Experiment

We use ILSVRC2012, a subset of ImageNet (Deng et al., 2009) which consists of about 1.28 million training images and 50000 validation images with 1000 different classes.

**Setting** We apply our method on pruning ResNet34 (traditional large architecture) (He et al., 2016), MobileNetV2 (efficient architecture) (Sandler et al., 2018) and MobileNetV3-small (an very small efficient architecture) (Howard et al., 2019) on ImageNet.

We use batch size 64 for both local and global imitation. We set the algorithm to converge when the gap between the cross entropy training loss before pruning and after pruning is smaller than $\epsilon$. We vary $\epsilon$ to get pruned model with different sizes. When conducting global imitation, we use the

| Model | Method | Top-1 Acc | Size (M) | FLOPs |
|---|---|---|---|---|
| ResNet34 | Full Model (He et al., 2016) | 73.4 | 21.8 | 3.68G |
| | $L_1$ norm (Li et al., 2017) | 72.1 | - | 2.79G |
| | Neural Imp (Molchanov et al., 2019) | 72.8 | - | 2.83G |
| | Rethink (Liu et al., 2018) | 72.9 | - | 2.79G |
| | More is Less (Dong et al., 2017) | 73.0 | - | 2.75G |
| | GFS (Ye et al., 2020) | 73.5 | 17.2 | 2.64G |
| | Ours | **73.5** | **14.9** | **2.20G** |
| | SPF (He et al., 2018a) | 71.8 | - | 2.17G |
| | FPGM (He et al., 2019) | 72.5 | - | 2.16G |
| | GFS (Ye et al., 2020) | 72.9 | 14.7 | 2.07G |
| | Ours | **73.3** | **13.5** | **1.90G** |
| MobileNetV2 | Full Model (Sandler et al., 2018) | 72.2 | 3.5 | 314M |
| | GFS (Ye et al., 2020) | 71.9 | 3.2 | 258M |
| | Ours | **72.2** | **3.2** | **245M** |
| | Uniform (Sandler et al., 2018) | 70.4 | 2.9 | 220M |
| | AMC (He et al., 2018b) | 70.8 | 2.9 | 220M |
| | Meta Pruning (Liu et al., 2019) | 71.2 | - | 217M |
| | LeGR (Chin et al., 2019) | 71.4 | - | 224M |
| | GFS (Ye et al., 2020) | 71.6 | 2.9 | 220M |
| | Ours | **71.7** | **2.9** | **218M** |
| | ThiNet (Luo et al., 2017) | 68.6 | - | 175M |
| | DPL (Zhuang et al., 2018) | 68.9 | - | 175M |
| | GFS (Ye et al., 2020) | 70.4 | 2.3 | 170M |
| | Ours | **70.5** | **2.3** | **170M** |
| MobileNetV3-Small | Full Model (Howard et al., 2019) | 67.5 | 2.5 | 64M |
| | Uniform (Howard et al., 2019) | 65.4 | 2.0 | 47M |
| | GFS (Ye et al., 2020) | 65.8 | 2.0 | 49M |
| | Ours | **66.4** | **2.0** | **48M** |

Table 2: Result on pruning deep neural networks on ImageNet.

Taylor approximation trick introduced in section 2.2.1 to accelerate global imitation. We start the approximation when the number of neurons is larger than 25 and we evaluate the top 5 neurons with largest $gr_{\boldsymbol{A},i}$ and pick up the best one to adjust. We find that this setting is able to produce the same pruning result as the exact version while substantially reduces the computation cost.

We finetune the pruned models with standard SGD optimizer with momentum 0.9 and weight decay $5 \times 10^{-5}$. All the pruned models are finetuned for 150 epochs with batch size 256 using cosine learning rate decay (Loshchilov & Hutter, 2016). We use initial learning 0.001 for ResNet34 and 0.01 for MobileNetV2 and MobileNetv3. We resize images to $224 \times 224$ resolution and adopt the standard data augmentation scheme (mirroring and shifting).

**Result** Table 2 reports the top1 accuracy, FLOPs and model size of the pruned network. Our algorithm consistently improves prior arts on network pruning.

## 3.3 DGCNN Experiment

We conduct experiment on the point cloud classification tasks on ModelNet40. Since the network structure used to extract the global information in point cloud usually requires to aggregate features from neighbor points, the high feature dimension heavily influence the forward time. We deploy our method on DGCNN. We compare with several baselines, including PointNet (Qi et al., 2017a), PointNet++ (Qi et al., 2017b), DGCNN with different width multipliers, and signed splitting steepest descent(Wu et al., 2020), which obtains a compact DGCNN by growing a extremely thin model. Table 3 shows that our method produces networks with comparable accuracy while with much less inference time. We refer readers to Appendix 5.7 for details of experiment settings.

| Model | Acc. | Forward time (ms) | # Param (M) |
|---|---|---|---|
| PointNet (Qi et al., 2017a) | 89.2 | 32.19 | 2.85 |
| PointNet++ (Qi et al., 2017b) | 90.7 | 331.4 | 0.86 |
| DGCNN (1.0x) | 92.6 | 60.12 | 1.81 |
| DGCNN (0.75x) | 92.4 | 48.06 | 1.64 |
| S3D (Wu et al., 2020) | **92.9** | 42.06 | 1.51 |
| Ours | **92.9** | **37.43** | 1.49 |
| DGCNN (0.5x) | 92.3 | 38.90 | 1.52 |
| Ours | **92.7** | **28.06** | 1.31 |
| DGCNN (0.25x) | 91.8 | 30.90 | 1.42 |
| Ours | **92.5** | **24.06** | 1.24 |

Table 3: Results on the ModelNet40 classification task.

## 4 Related Work

**Greedy Method**   Our method is highly related to Ye et al. (2020), which is also a greedy method with $\mathcal{O}(n^{-2})$ error rate for pruning over-parameterized two layer network. In comparison, we obtain exponential decay rate for pruning deep neural network with no requirement on the over-parameterization of full model. Our local imitation method is also related to Frank Wolfe algorithm (Frank & Wolfe, 1956). Compared with it, our local imitation is a bi-level greedy joint optimization method while Frank Wolfe first searches for best direction and then conduct descent greedily.

**Theory on Lottery Ticket Hypothesis**   (Malach et al., 2020) aims to prove the existence of sub-network inside a random network that well approximates an unknown target network which has finite width and depth. It shows that a sufficiently large random network (with a specific structure) contains such a subnetwork with width of higher order compared with the targeted network. Later Pensia et al. (2020); Orseau et al. (2020) improve the result by reducing the size of the original random network. Elesedy et al. (2020) also gives analysis of Lottery Ticket Hypothesis in linear model using the tool from compressive sensing. Compared with our method, their theoretical results require strong structure assumptions on the full model and pruned model. Besides, they fail to give an efficient algorithm to search for the subnetwork for deep learning model in practice. Notice that Ye et al. (2020) also gives analysis on Lottery Ticket Hypothesis and it is straightforward to combine their framework and our analysis to give faster rate.

**Structured Pruning**   Existing methods on structured pruning includes the sparsity regularization based training methods, e.g., Molchanov et al. (2017a); Liu et al. (2017); Ye et al. (2018); Huang & Wang (2018); criterion based methods, e.g., Molchanov et al. (2017b); Li et al. (2017); Molchanov et al. (2019), reconstruction error based method, e.g., He et al. (2017); Luo et al. (2017); Zhuang et al. (2018); Yu et al. (2018) and direct search method, e.g., He et al. (2018b); Liu et al. (2019). Our local imitation falls into the class of reconstruction error based method. Compared with those existing works, the proposed greedy optimization method enjoys good convergence property under weak assumptions and achieve better practical performance. Zhou et al. (2020) proposes a layer-wise imitation based training method for training deep and thin network, which is able to reduce the optimization error caused by the depth of the network. Their work is orthogonal to our work as we focus on reducing the error caused by small width.

## 5 Conclusion

This paper proposes a greedy optimization based pruning method, which is guaranteed to find a set of winning tickets (neurons) that approximates the fully trained unpruned network with exponential decay error rate w.r.t the number of selected tickets. The proposed pruning method is efficient with small time and space complexity and can be generally applied to various modern deep learning models.

**Broader Impact Statement**   This work proposes a greedy optimization based pruning method, which has strong theoretical guarantee and good empirical performance. It gives positive improvement to the community of network efficiency. Our work do not have any negative societal impacts that we can foresee in the future.

**Acknowledgement**   This paper is supported in part by NSF CAREER 1846421, SenSE 2037267 and EAGER 2041327.

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
