[Supplementary Material]

# Appendix

We introduce the following extra notations which are used in several parts of the Appendix. Suppose that $\boldsymbol{\theta}_i^\ell, i \in [N]$ is the weight of the $N$ neurons of the $\ell$-th layer in the original network $F$. We simplify the notation by denoting $\boldsymbol{\theta}_i^\ell = \boldsymbol{\theta}_i, i \in [N]$. Suppose $\sigma(\boldsymbol{\theta}, \boldsymbol{z}) \in \mathbb{R}^d$, we define the data-dependent feature by

$$\phi_{\text{Matrix}}(\boldsymbol{\theta}) = \left[ \sigma(\boldsymbol{\theta}, \boldsymbol{z}^{(1)}), ..., \sigma(\boldsymbol{\theta}, \boldsymbol{z}^{(m)}) \right]^\top \in \mathbb{R}^{m \times d},$$

and its vectorization

$$\phi(\boldsymbol{\theta}) = \left[ \text{vec}^\top \left( \sigma(\boldsymbol{\theta}, \boldsymbol{z}^{(1)}) \right), ..., \text{vec}^\top \left( \sigma(\boldsymbol{\theta}, \boldsymbol{z}^{(m)}) \right) \right]^\top \in \mathbb{R}^{md}.$$

We also define $\boldsymbol{h}_i = \phi(\boldsymbol{\theta}_i), i \in [N], \bar{\boldsymbol{h}} = \frac{1}{N} \sum_{i=1}^N \boldsymbol{h}_i$ and $\boldsymbol{h}_{\boldsymbol{A}} = \sum_{i=1}^N a_i h_i$ with $\boldsymbol{A} = [a_1, a_2, ..., a_N]$. Define $\mathcal{M} = \text{conv} \left( \{ \phi(\boldsymbol{\theta}_i) \mid i \in [N] \} \right)$ as the convex hull generated by the set $\{ \phi(\boldsymbol{\theta}_i) \mid i \in [N] \}$. Given some set $M \subseteq \mathbb{R}^d$, we denote the relative interior of $M$ by ri$M$, the closure of $M$ by cl$M$ and the affine hull of $M$ by Aff$M$. We define $\mathcal{B}(\boldsymbol{x}_0, r)$ as the ball centered at $\boldsymbol{x}_0$ with radius $r$.

## 5.1 Details on Solving (4) for Local Imitation

Now we describe the approach to solve problem (4) with only compute one forward pass. Define

$$V_{i,\boldsymbol{A}}(\gamma) = \bar{\mathbb{D}}[(1 - \gamma) f_{\ell, \boldsymbol{A}} + \gamma \sigma(\boldsymbol{\theta}_i^\ell, \cdot), F_\ell] = \gamma^2 g_{i,\boldsymbol{A}} - 2\gamma q_{i,\boldsymbol{A}} + \bar{\mathbb{D}}[f_{\ell, \boldsymbol{A}}, F_\ell],$$

where $q_{i,\boldsymbol{A}} = \mathbb{E}_{\boldsymbol{z} \sim \mathcal{D}_m^\ell} \left[ (F_\ell - f_{\ell, \boldsymbol{A}}) \left( \sigma(\boldsymbol{\theta}_i^\ell, \cdot) - f_{\ell, \boldsymbol{A}} \right) \right], g_{i,\boldsymbol{A}} = \mathbb{E}_{\boldsymbol{z} \sim \mathcal{D}_m^\ell} \left[ \left( \sigma(\boldsymbol{\theta}_i^\ell, \cdot) - f_{\ell, \boldsymbol{A}} \right)^2 \right].$

Notice that it is easy to obtain $\sigma(\boldsymbol{\theta}_i^\ell, \boldsymbol{z}^{(j)})$ for all $i \in [N]$ and $j \in [m]$ by feeding the dataset into the neural network once, as it is the output of neuron $i$ in layer $j$. And thus $q_{i,\boldsymbol{A}(k)}$ and $g_{i,\boldsymbol{A}(k)}$ can be calculated cheaply given $\sigma(\boldsymbol{\theta}_i^\ell, \boldsymbol{z}^{(j)}), i \in [N]$ and $j \in [m]$. Define $\tilde{\gamma}_{i,\boldsymbol{A}(k)} = q_{i,\boldsymbol{A}(k)}/g_{i,\boldsymbol{A}(k)}$, which is the optimum of $V_{i,\boldsymbol{A}(k)}(\gamma)$ w.r.t. $\gamma$ given $i$ (here the optimization of $\gamma$ is unconstrained). The following theorem shows some properties of the greedy local imitation method.

**Theorem 3** *Under assumption 1, if $\bar{\mathbb{D}}[f_{\ell, \boldsymbol{A}(k)}, F_\ell] > 0$, then we have $\tilde{\gamma}_{\ell, i_{\ell,k}^*} < 1$.*

Now we proceed to show how to obtain $i_k^*$ and $\gamma_k^*$ efficiently. Suppose at iteration $k$, $\bar{\mathbb{D}}[f_{\ell, \boldsymbol{A}(k)}, F_\ell] > 0$ (otherwise the algorithm has converged). Given some neuron $i$ with $a_i(k) = 0$, we first calculate $\tilde{\gamma}_{i,\boldsymbol{A}(k)}$. If $\tilde{\gamma}_{i,\boldsymbol{A}(k)} \in (0, 1)$, then we define the score of this neuron by the decrease of loss with this neuron selected, i.e.,

$$\texttt{score}_{\boldsymbol{A}(k)}(i) := \bar{\mathbb{D}}[f_{\ell, \boldsymbol{A}}, F_\ell] - \min_{\gamma \in U_i} V_{i,\boldsymbol{A}(k)}(\gamma) = -q_{i,\boldsymbol{A}(k)}^2/g_{i,\boldsymbol{A}(k)}.$$

If $\tilde{\gamma}_{i,\boldsymbol{A}(k)} \geq 1$, then from Theorem 3, we know that $i \neq i_k^*$. If $\tilde{\gamma}_{i,\boldsymbol{A}(k)} \leq 0$ and if $i = i_k^*$, we have $\gamma_k^* = 0$, which implies that $\bar{\mathbb{D}}[f_{\ell, \boldsymbol{A}(k+1)}, F_\ell] = \bar{\mathbb{D}}[f_{\ell, \boldsymbol{A}(k)}, F_\ell]$. It makes contradiction to Theorem 3, which implies that $i \neq i_k^*$. In this two cases, since neuron $i$ is not the optimal neuron to select, we can safely set $\texttt{score}_{\boldsymbol{A}(k)}(i) = 0$. For neuron $i$ with $a_i(k) > 0$. Similarly, if $\tilde{\gamma}_{i,\boldsymbol{A}(k)} \geq 1$, then $i \neq i_k^*$ and thus we set $\texttt{score}_{\boldsymbol{A}(k)}(i) = 0$. If $\tilde{\gamma}_{i,\boldsymbol{A}(k)} \in U_i$, then similarly $\texttt{score}_{\boldsymbol{A}(k)}(i) = -q_{i,\boldsymbol{A}(k)}^2/g_{i,\boldsymbol{A}(k)}$. If $\tilde{\gamma}_{i,\boldsymbol{A}(k)} < -a_i(k)/(1 - a_i(k))$, then

$$\texttt{score}_{\boldsymbol{A}(k)}(i) = V_{i,\boldsymbol{A}(k)}(-a_i(k)/(1 - a_i(k))).$$

And thus we have $i_k^* = \arg\max_{i \in [N]} \texttt{score}_{\boldsymbol{A}(k)}(i)$. Notice that the score of most neuron can be calculated cheaply using $q_{i,\boldsymbol{A}(k)}$ and $g_{i,\boldsymbol{A}(k)}$. The only exception are neuron with $a_i(k) > 0$ and $\tilde{\gamma}_{i,\boldsymbol{A}(k)} < -a_i(k)/(1 - a_i(k))$. However, its score can be calculated using $\sigma(\boldsymbol{\theta}_i^\ell, \cdot)$ and thus no extra forward pass is required.

## 5.2 Local Imitation with Fixed Step Sizes

In this section we give detailed discussion on local imitation with a fixed step size scheme shown in Section 2.2.1. Different from the greedy optimization (4), in this scheme, as the step size is fixed, the solution returned in each iteration is no better than the one of (4). As a consequence, it gives slower convergence rate.

**Theorem 4** *Under Assumption 1, at each step $k$ of the greedy optimization in 5, we have*
$$\mathbb{D}[f_{\boldsymbol{A}(k)}, F] \le \|H\|_{Lip}^2 \, \bar{\mathbb{D}}\left[f_{\ell, \boldsymbol{A}(k)}, F_\ell\right] = \mathcal{O}((k+1)^{-2}) \text{ and } \|\boldsymbol{A}(k)\|_0 \le k+1.$$

## 5.3 Theory on Greedy Global Imitation

Now we give the theoretical result on greedy global imitation. Denote $\kappa_1 = \sup_{\boldsymbol{A} \in \Omega_N} \frac{\|H \circ \bar{\boldsymbol{h}} - H \circ \boldsymbol{h}_{\boldsymbol{A}}\|}{\|\bar{\boldsymbol{h}} - \circ \boldsymbol{h}_{\boldsymbol{A}}\|}$, $\kappa_2 = \sup_{\boldsymbol{A} \in \Omega_N} \frac{\|\bar{\boldsymbol{h}} - \circ \boldsymbol{h}_{\boldsymbol{A}}\|}{\|H \circ \bar{\boldsymbol{h}} - H \circ \boldsymbol{h}_{\boldsymbol{A}}\|}$ and $D$ as the diameter of $\mathcal{M}$, which is defined in Lemma 5. Notice that $\kappa_1 \kappa_2 \ge 1$. Using Lemma 3, we know that $\bar{\boldsymbol{h}} \in \text{ri}\mathcal{M}$, which indicate that there exists some $\lambda > 0$ such that

$$\mathcal{B}(\bar{\boldsymbol{h}}, \lambda) \cap \text{Aff}\mathcal{M} \subseteq \mathcal{M},$$

where $\mathcal{B}(\bar{\boldsymbol{h}}, \lambda)$ denotes the ball with radius $\lambda$ centered at $\bar{\boldsymbol{h}}$.

**Theorem 5 (Complete Version of Theorem 2)** *Suppose Assumption 1 holds. Further suppose that 1. $D^2 \ge \kappa_1^2 \kappa_2^2 (D^2 - \lambda^2)$; 2. at initialization $\left\|\bar{\boldsymbol{h}} - \boldsymbol{h}_{\boldsymbol{A}(0)}\right\| \le R$; 3. $\kappa_1 D \le R$, where we define $R = \frac{\kappa_1^2 \kappa_2 \lambda + \kappa_1 \sqrt{\kappa_1^2 \kappa_2^2 (\lambda^2 - D^2) + D^2}}{(\kappa_1^2 \kappa_2^2 - 1)}$ ($R = +\infty$ if $\kappa_1 \kappa_2 = 1$). Then we have $\mathbb{D}[f_{\boldsymbol{A}(k)}, F] = \mathcal{O}((k+1)^{-2})$, and $\|\boldsymbol{A}(k)\|_0 \le k+1$.*

**Remark** Here the descending property of global imitation is influenced by the non-linear mapping $H$. As a consequence, the algorithm gives good convergence property when the whole dynamics is guaranteed to stay in a proper convergence region ($R$). The first extra assumption assumes the existence of this convergence region; The second extra assumption assumes a good initialization to ensure the dynamics stays in the convergence region at initialization; The third assumption can be roughly interpreted as assuming the dynamics will not jump out of the convergence region during descending. Notice that the extra assumptions holds when $\kappa_1$ and $\kappa_2$ is sufficiently close to 1.

## 5.4 Details on Taylor Approximation Tricks

In this section we give details on the computation of Taylor approximation tricks. Notice that

$$
\begin{aligned}
gr_{\boldsymbol{A},i} =& \frac{\partial}{\partial \gamma} \mathbb{D}\left[H \circ \left[(1-\gamma)f_{\ell,\boldsymbol{A}} + \gamma \sigma(\boldsymbol{\theta}_i, \cdot)\right], F\right]\bigg|_{\gamma=0} \\
=& \frac{\partial}{\partial \gamma} \mathbb{E}_{\boldsymbol{z} \sim \mathcal{D}_m^\ell}\left(H \circ \left[(1-\gamma)f_{\ell,\boldsymbol{A}}(\boldsymbol{z}) + \gamma \sigma(\boldsymbol{\theta}_i, \boldsymbol{z})\right] - H \circ F_\ell(\boldsymbol{z})\right)^2\bigg|_{\gamma=0} \\
=& 2\mathbb{E}_{\boldsymbol{z} \sim \mathcal{D}_m^\ell}\left(H \circ \left[(1-\gamma)f_{\ell,\boldsymbol{A}}(\boldsymbol{z}) + \gamma \sigma(\boldsymbol{\theta}_i, \boldsymbol{z})\right] - H \circ F_\ell(\boldsymbol{z})\right) \frac{\partial}{\partial \gamma}\left(H \circ \left[(1-\gamma)f_{\ell,\boldsymbol{A}}(\boldsymbol{z}) + \gamma \sigma(\boldsymbol{\theta}_i, \boldsymbol{z})\right]\right)\bigg|_{\gamma=0} \\
=& 2\mathbb{E}_{\boldsymbol{z} \sim \mathcal{D}_m^\ell}\left(H \circ f_{\ell,\boldsymbol{A}}(\boldsymbol{z}) - H \circ F_\ell(\boldsymbol{z})\right) H'\left(f_{\ell,\boldsymbol{A}}(\boldsymbol{z})\right)\left(\sigma(\boldsymbol{\theta}_i, \boldsymbol{z}) - f_{\ell,\boldsymbol{A}}(\boldsymbol{z})\right) \\
=& 2\mathbb{E}_{\boldsymbol{z} \sim \mathcal{D}_m^\ell}\left(H \circ f_{\ell,\boldsymbol{A}}(\boldsymbol{z}) - H \circ F_\ell(\boldsymbol{z})\right) H'\left(f_{\ell,\boldsymbol{A}}(\boldsymbol{z})\right)\left(\sigma(\boldsymbol{\theta}_i, \boldsymbol{z}) - \sum_{j=1}^N a_j \sigma(\boldsymbol{\theta}_j, \boldsymbol{z})\right) \\
=& 2\sum_{j=1}^N \left(\mathbb{I}\{j=i\} - a_j\right) r_{\boldsymbol{A},j}.
\end{aligned}
$$

Thus the key quantities we want to obtain is

$$r_{\boldsymbol{A},i} = \mathbb{E}_{\boldsymbol{z} \sim \mathcal{D}_m^\ell}\left[\left(H \circ f_{\ell,\boldsymbol{A}}(\boldsymbol{z}) - H \circ F_\ell(\boldsymbol{z})\right) H'(f_{\ell,\boldsymbol{A}}(\boldsymbol{z})) \sigma(\boldsymbol{\theta}_i^\ell, \boldsymbol{z})\right].$$

And once we obtain $r_{\boldsymbol{A},i}$, we are able to calculate $gr_{\boldsymbol{A},i} = 2\sum_{j=1}^{N}\left(\mathbb{I}\{j=i\}-a_j\right)r_{\boldsymbol{A},j}$. Now we introduce how to calculate $r_{\boldsymbol{A},i}$ efficiently by introducing an ancillary variable. Suppose when pruning layer $\ell$, we have

$$f_{\ell,\boldsymbol{A}}(\boldsymbol{z}) = \sum_{i=1}^{N}a_i\sigma(\boldsymbol{\theta}_i^{\ell},\boldsymbol{z}) = \sum_{i=1}^{N}(a_i+b_i)\sigma(\boldsymbol{\theta}_i^{\ell},\boldsymbol{z}), \ \ \text{where } b_i = 0 \ \forall i \in [N].$$

Here $b_i$ is the introduced ancillary variable, which alway takes 0 value. We have

$$\frac{\partial}{\partial b_i}\mathbb{E}_{\boldsymbol{z}\sim\mathcal{D}_m^{\ell}}\left(H\circ\left(\sum_{i=1}^{N}(a_i+b_i)\sigma(\boldsymbol{\theta}_i^{\ell},\boldsymbol{z})\right) - H\circ F_{\ell}(\boldsymbol{z})\right)^2\Bigg|_{b_i=0}$$

$$=\mathbb{E}_{\boldsymbol{z}\sim\mathcal{D}_m^{\ell}}\left(H\circ f_{\ell,\boldsymbol{A}}(\boldsymbol{z}) - H\circ F_{\ell}(\boldsymbol{z})\right)\frac{\partial}{\partial b_i}\left(H\circ f_{\ell,\boldsymbol{A}}\right)\Bigg|_{b_i=0}$$

$$=\mathbb{E}_{\boldsymbol{z}\sim\mathcal{D}_m^{\ell}}\left(H\circ f_{\ell,\boldsymbol{A}}(\boldsymbol{z}) - H\circ F_{\ell}(\boldsymbol{z})\right)H'(f_{\ell,\boldsymbol{A}})\frac{\partial}{\partial b_i}f_{\ell,\boldsymbol{A}}\Bigg|_{b_i=0}$$

$$=\mathbb{E}_{\boldsymbol{z}\sim\mathcal{D}_m^{\ell}}\left(H\circ f_{\ell,\boldsymbol{A}}(\boldsymbol{z}) - H\circ F_{\ell}(\boldsymbol{z})\right)H'(f_{\ell,\boldsymbol{A}})\frac{\partial}{\partial b_i}\left(\sum_{i=1}^{N}(a_i+b_i)\sigma(\boldsymbol{\theta}_i^{\ell},\boldsymbol{z})\right)\Bigg|_{b_i=0}$$

$$=\mathbb{E}_{\boldsymbol{z}\sim\mathcal{D}_m^{\ell}}\left(H\circ f_{\ell,\boldsymbol{A}}(\boldsymbol{z}) - H\circ F_{\ell}(\boldsymbol{z})\right)H'(f_{\ell,\boldsymbol{A}})\sigma(\boldsymbol{\theta}_i^{\ell},\boldsymbol{z})$$

$$=r_{\boldsymbol{A},i}.$$

This for implementation in practice, we can introduce $b_i$ with its value fixed 0 and calculate its gradient using, which is $r_{\boldsymbol{A},i}$ using the auto differentiate operator in common deep learning libraries.

## 5.5 Details on Numeric Verification of Rate

In this section, we give details on the toy experiment on verifying numeric rate. We first introduce the problem setup for the comparison between pruning and direct gradient training in obtaining small network. We use the two-hidden-layer deep mean field network formulated by Araújo et al. (2019). Suppose that the second hidden layer (the one close to output) has 50 neurons; the first hidden layer (the one close to input) has $n \leq 50$ neurons with 50 dimensional feature map; and the input has 100 dimension. That is, we consider the following deep mean field network

$$F^n(\boldsymbol{x}) = F_2 \circ F_1^n(\boldsymbol{x}),$$

where

$$F_1^n(\boldsymbol{x}) = \frac{1}{n}\sum_{i=1}^{n}a_{1,i}\text{ReLU}(\boldsymbol{b}_{1,i}^{\top}\boldsymbol{x})$$

with $\boldsymbol{x} \in \mathbb{R}^{100}$, $\boldsymbol{b}_{1,i} \in \mathbb{R}^{100\times50}$, $a_{1,i} \in \mathbb{R}$. And

$$F_2(\boldsymbol{z}) = \frac{1}{50}\sum_{i=1}^{50}a_{2,i}\text{ReLU}(\boldsymbol{b}_{2,i}^{\top}\boldsymbol{z}),$$

with $\boldsymbol{z} \in \mathbb{R}^{50}$, $\boldsymbol{b}_{2,i} \in \mathbb{R}^{50}$, $a_{1,i} \in \mathbb{R}$. Suppose that we train the original network $F^N$ with $N = 50$ neuron at the first hidden layer using gradient descent defined in Araújo et al. (2019) for $T$ time ($T < \infty$) with random initialization. To obtain a small network with $n$ neurons at the first hidden layer, we consider two approaches. In the first approach, we prune the first hidden layer of the trained $F^N$ using local imitation to obtain $F_{\text{local}}^n$ where $n$ indicates the number of neurons remained in the first hidden layer. In the second approach, we direct train the network $F_{\text{direct train}}^n$ with $n$ neurons in the first layer using the same gradient descent dynamics, initialization and training time as that in training $F^N$. By the analysis in Araújo et al. (2019), we have $\mathbb{D}[F_{\text{direct train}}^n, F^N] = \mathcal{O}(n^{-1})$. And by Theorem 1, we have $\mathbb{D}[F_{\text{local}}^n, F^N] = \mathcal{O}(\exp(-\lambda n))$ for some $\lambda > 0$. This implies that pruning is provably much better than directly training in obtaining compact neural network.

Now we introduce the experiment settings. To simulate the data, we first generate a random network

$$F_{\text{gen}}(\boldsymbol{x}) = (\exp(\boldsymbol{w}_2/10) - 0.5)^{\top}\text{Tanh}(\sin(2\pi\boldsymbol{w}_1)^{\top}\boldsymbol{x}/5)/1000,$$

where $\boldsymbol{w}_1 \in \mathbb{R}^{1000 \times 100}$ and $\boldsymbol{w}_2 \in \mathbb{R}^{1000}$ is generated by randomly sampling from uniform distribution Unif$[0, 1]$ (each element is sampled independently). And then we generate the training data by sampling feature $\boldsymbol{x}$ from Unif$[0, 1]$ (each coordinate is sampled independently) and then generate label $y = F_{\text{gen}}(\boldsymbol{x})$. The simulated training dataset consists of 200 data points. We initialize the parameters of $F^N$ and $F^n_{\text{direct train}}$ from standard Gaussian distribution with variance 1, $\mathcal{N}(0, 1)$ (each element are initialized independently) and both $F^N$ and $F^n_{\text{direct train}}$ are trained using the same and sufficiently long time to ensure convergence. We also include the pruned model using global imitation, which is denoted as $F^n_{\text{global}}$. The pruned models are not finetuned. We vary different $n$ and summarize the discrepancy.

## 5.6 Theory on Pruning All Layers

In the main text, we mainly discuss the convergence rate of pruning one layer. In this section, we discuss how to apply our convergence rate for single layer pruning to obtain an overall convergence rate. Following the layer-wise procedure introduced in Section 2.4, suppose that the algorithms prunes $F_\ell$ to $f_{\ell, \boldsymbol{A}_\ell}$, $\ell \in [L]$. And thus, during the layer-wise pruning, the algorithm generates a sequence of pruned networks

$$f_{[0]} = F_L \circ F_{L-1} \circ ... \circ F_3 \circ F_2 \circ F_1$$
$$f_{[1]} = F_L \circ F_{L-1} \circ ... \circ F_3 \circ F_2 \circ f_{1, \boldsymbol{A}_1}$$
$$f_{[2]} = F_L \circ F_{L-1} \circ ... \circ F_3 \circ f_{2, \boldsymbol{A}_2} \circ f_{1, \boldsymbol{A}_1}$$
$$\vdots$$
$$f_{[L-1]} = F_L \circ f_{L-1, \boldsymbol{A}_{L-1}} \circ ... \circ f_{3, \boldsymbol{A}_3} \circ f_{2, \boldsymbol{A}_2} \circ f_{1, \boldsymbol{A}_1}$$
$$f_{[L]} = f_{L, \boldsymbol{A}_L} \circ f_{L-1, \boldsymbol{A}_{L-1}} \circ ... \circ f_{3, \boldsymbol{A}_3} \circ f_{2, \boldsymbol{A}_2} \circ f_{1, \boldsymbol{A}_1}$$

Thus here $f_{[\ell]}$ is the network with the first $\ell$ layers pruned, $f_{[L]}$ is the final pruned network with all layers pruned and $f_{[0]}$ is the original network. Notice that $f_{[\ell]}$ is obtained by pruning the $\ell$-th layer of $f_{[\ell-1]}$. In this step, we suppose that we try both greedy local and global imitation and obtain $f_{\ell, \boldsymbol{A}_\ell^{\text{local}}}$ and $f_{\ell, \boldsymbol{A}_\ell^{\text{global}}}$ with $\left\| \boldsymbol{A}_\ell^{\text{local}} \right\|_0 = \left\| \boldsymbol{A}_\ell^{\text{global}} \right\|_0$. And if $\mathbb{D}[F_L \circ ... F_{\ell+1} \circ f_{\ell, \boldsymbol{A}_\ell^{\text{local}}} \circ ... \circ f_{1, \boldsymbol{A}_1}, f_{[\ell-1]}] \le \mathbb{D}[F_L \circ ... F_{\ell+1} \circ f_{\ell, \boldsymbol{A}_\ell^{\text{global}}} \circ ... \circ f_{1, \boldsymbol{A}_1}, f_{[\ell-1]}]$, we set $\boldsymbol{A}_\ell = \boldsymbol{A}_\ell^{\text{local}}$, else we set $\boldsymbol{A}_\ell = \boldsymbol{A}_\ell^{\text{global}}$. Define $H_{[\ell]} = F_L \circ F_{L-1} \circ ... \circ F_{\ell+1}$, $\ell \in [L-1]$ (here $H_{[L-1]} = F_L$) and $\boldsymbol{z}_{[\ell]}^{(i)} = f_{\ell-1, \boldsymbol{A}_{\ell-1}} \circ ... \circ f_{1, \boldsymbol{A}_1}(\boldsymbol{x}^{(i)})$, $\ell \in [L-1]$ (here we define $\boldsymbol{z}_{[1]}^{(i)} = \boldsymbol{x}^{(i)}$). The set $\mathcal{D}_m^{[\ell]} := \left( \boldsymbol{z}_{[\ell]}^{(i)} \right)_{i=1}^m$ denotes the distribution of training data pushed through the first $\ell - 1$ layers.

We introduce the following assumption on the boundedness.

**Assumption 2** *Assume that for any $i \in [N]$, $\ell \in [L-1]$, $\boldsymbol{z}_{[\ell]}^{(j)} \in \mathcal{D}_m^{[\ell]}$, we have $\left\| \sigma(\boldsymbol{\theta}_i^\ell, \boldsymbol{z}_{[\ell]}^{(j)}) \right\| \le c_2$ and $\left\| H_{[\ell]} \right\|_{Lip} \le c_2$ for some $c_2 < \infty$.*

**Theorem 6 (Overall Convergence)** *Under assumption 2, we have* $\sqrt{\mathbb{D}[f_{[L]}, F]} = \mathcal{O}\left( \sum_{\ell=1}^L \exp\left( -\frac{\lambda_\ell}{2} \left\| \boldsymbol{A}_\ell \right\|_0 \right) \right)$, *with $\lambda_\ell > 0$ for all $\ell \in [L]$ depending on $f_{[\ell-1]}$.*

## 5.7 DGCNN Experiment

We deploy our method on DGCNN (Wang et al., 2019). DGCNN contains 4 EdgeConv layers that use K-Nearest-Neighbor(KNN) to aggregate the information from the output of convolution operation. Pruning the convolution operation in EdgeConv can significantly speed up the KNN operation and therefor make the whole model more computational efficient.

**Settings** The full network is trained with SGD optimizer with momentum 0.9 and weight decay $1 \times 10^{-4}$. We train the model using 64 batch size with an initial learning rate 0.1 for 250 epochs. We apply cosine learning rate scheduler during the training and decrease the learning rate to 0.001 at the final epoch. During the pruning, we use 32 batch sizes and the other settings keep the same as our ImageNet experiment in Section 3.2.

## Technical Lemmas

We introduce several technical Lemmas that are useful for proving the main theorems.

**Lemma 1** *Given some convex set $M \subset \mathbb{R}^d$, for any $\boldsymbol{q}_1 \in riM$ and $\boldsymbol{q}_2 \in clM$. Then all the points from the half-segment $[\boldsymbol{q}_1, \boldsymbol{q}_2)$ belongs to the relative interior of $M$, i.e.,*

$$[\boldsymbol{q}_1, \boldsymbol{q}_2) \hat{=} \{(1-\lambda)\boldsymbol{q}_1 + \lambda\boldsymbol{q}_2 \mid 0 \leq \lambda < 1\} \subseteq riM.$$

**Lemma 2** *Let $M$ be a convex set in $\mathbb{R}^d$, then if $M$ is nonempty, then the relative interior of $M$ is nonempty.*

Lemma 1 and 2 are classic results from convex optimization.

**Lemma 3** *Define*

$$M = conv\{\boldsymbol{q} \mid \boldsymbol{q} \in S\},$$

*where $S = \{\boldsymbol{q}_1, ..., \boldsymbol{q}_n\} \subseteq \mathbb{R}^d$ with $1 \leq n < \infty$. Define $\bar{\boldsymbol{q}} = \frac{1}{n}\sum_{i=1}^{n} \boldsymbol{q}_i$, then $\bar{\boldsymbol{q}} \in riM$.*

**Lemma 4** *Suppose that for some $\lambda > 0$ such that $\left(\mathcal{B}(\bar{\boldsymbol{h}}, \lambda) \cap Aff\mathcal{M}\right) \subseteq \mathcal{M}$, then $\max_{\boldsymbol{s}\in\mathcal{M}} \langle \bar{\boldsymbol{h}} - \boldsymbol{h}_A, \bar{\boldsymbol{h}} - \boldsymbol{s} \rangle \geq \lambda \|\bar{\boldsymbol{h}} - \boldsymbol{h}_A\|$.*

**Lemma 5** *Under Assumption 1, for any $\boldsymbol{h}, \boldsymbol{h}' \in \mathcal{M}$, $\|\boldsymbol{h} - \boldsymbol{h}'\| \leq D$ for some $D \leq 2\sqrt{m}c_1$. Here $D$ can be viewed as the diameter of $\mathcal{M}$.*

**Lemma 6** *Under assumption 1, suppose $\tilde{\boldsymbol{s}}_k^* = \underset{\boldsymbol{s}\in\mathcal{M}}{\arg\min} \langle \bar{\boldsymbol{h}} - \boldsymbol{h}_{\boldsymbol{A}(k)}, \boldsymbol{s} - \boldsymbol{h}_{\boldsymbol{A}(k)} \rangle$ and $\tilde{\gamma}_k^* = \underset{\gamma\in[0,1]}{\arg\min} \left\|\boldsymbol{h}_{\boldsymbol{A}(k)} + \gamma\left(\tilde{\boldsymbol{s}}_k^* - \boldsymbol{h}_{\boldsymbol{A}(k)}\right) - \bar{\boldsymbol{h}}\right\|^2$, then*

$$\left\|\boldsymbol{h}_{\boldsymbol{A}(k)} + \tilde{\gamma}_k^*\left(\tilde{\boldsymbol{s}}_k^* - \boldsymbol{h}_{\boldsymbol{A}(k)}\right) - \bar{\boldsymbol{h}}\right\|^2 \leq \rho \left\|\boldsymbol{h}_{A(k)} - \bar{h}\right\|^2,$$

*for some $\rho \in (0,1)$.*

**Lemma 7** *Consider the following number sequence $x_{k+1}^2 \leq ax_k^2 - bx_k + c$. Suppose that this number sequence satisfies the following conditions: (1) $a > 1$, $b \geq 0$, $c \geq 0$; (2) $x_k \geq 0$ for any $k$; (3) $(a-1)x^2 - bx^2 + c$ has two real roots $z_1 \leq z_2$; (we allow $z_1 = z_2$); (4) $\sqrt{c} \leq z_2$; (5) $x_0 \leq z_2$. Then $\sup_k x_k \leq z_2$.*

## Proof of Main Theorems

### 5.7.1 Proof of Theorem 1

Using Lemma 3, we know that $\bar{\boldsymbol{h}} \in ri\mathcal{M}$, which indicate that there exists some $\lambda > 0$ such that

$$\mathcal{B}(\bar{\boldsymbol{h}}, \lambda) \cap Aff\mathcal{M} \subseteq \mathcal{M},$$

where $\mathcal{B}(\bar{\boldsymbol{h}}, \lambda)$ denotes the ball with radius $\lambda$ centered at $\bar{\boldsymbol{h}}$. Define $Extre(\mathcal{M})$ as the set of extreme points of $\mathcal{M}$, we know that $Extre(\mathcal{M}) \subseteq \{\boldsymbol{h}_1, ..., \boldsymbol{h}_N\}$. Consider the following problem

$$\min_{\boldsymbol{s}\in\mathcal{M}} \langle \bar{\boldsymbol{h}} - \boldsymbol{h}_{\boldsymbol{A}(k)}, \boldsymbol{s} - \boldsymbol{h}_{\boldsymbol{A}(k)} \rangle.$$

As the objective $\langle \bar{\boldsymbol{h}} - \boldsymbol{h}_{\boldsymbol{A}(k)}, \boldsymbol{s} - \boldsymbol{h}_{\boldsymbol{A}(k)} \rangle$ is linear w.r.t. $\boldsymbol{s}$, we know that $\boldsymbol{s} \in Extre(\mathcal{M}) \subseteq \{\boldsymbol{h}_1, ..., \boldsymbol{h}_N\}$. Also, for any $i \in [N]$, we have $[0,1] \subseteq U_i$. This gives that

$$\min_{i\in[N]} \min_{\gamma\in U_i} \bar{\mathbb{D}}[(1-\gamma)f_{\ell,\boldsymbol{A}(k)} + \gamma\sigma(\boldsymbol{\theta}_i, \cdot), F_\ell] \leq \min_{\gamma\in[0,1]} \left\|\boldsymbol{h}_{\boldsymbol{A}(k)} + \gamma\left(\tilde{\boldsymbol{s}}_k^* - \boldsymbol{h}_{\boldsymbol{A}(k)}\right) - \bar{\boldsymbol{h}}\right\|^2$$

$$\leq (1 - \lambda^2/D^2) \left\|\boldsymbol{h}_{\boldsymbol{A}(k)} - \bar{\boldsymbol{h}}\right\|^2,$$

where $\tilde{\boldsymbol{s}}_k^* = \arg\min_{\boldsymbol{s}\in\mathcal{M}} \langle \bar{\boldsymbol{h}} - \boldsymbol{h}_{\boldsymbol{A}(k)}, \boldsymbol{s} - \boldsymbol{h}_{\boldsymbol{A}(k)} \rangle$. Here the last inequality is by Lemma 6. This gives that

$$\left\|\boldsymbol{h}_{\boldsymbol{A}(k+1)} - \bar{\boldsymbol{h}}\right\|^2 \leq (1 - \lambda^2/D^2) \left\|\boldsymbol{h}_{\boldsymbol{A}(k)} - \bar{\boldsymbol{h}}\right\|^2.$$

And thus we have
$$\left\| \boldsymbol{h}_{\boldsymbol{A}(k)} - \bar{\boldsymbol{h}} \right\|^2 \le (1 - \lambda^2/D^2)^k \left\| \boldsymbol{h}_{\boldsymbol{A}(0)} - \bar{\boldsymbol{h}} \right\|^2.$$

And thus we have
$$\mathbb{D}[f_{\boldsymbol{A}(k)}, F] \le c_1^2 (1 - \lambda^2/D^2)^k \left\| \boldsymbol{h}_{\boldsymbol{A}(0)} - \bar{\boldsymbol{h}} \right\|^2 \le c_1^2 (1 - \lambda^2/D^2)^{\|\boldsymbol{A}(k)\|_0} \left\| \boldsymbol{h}_{\boldsymbol{A}(0)} - \bar{\boldsymbol{h}} \right\|^2,$$

where the last inequality is by $\|\boldsymbol{A}(k)\|_0 \le k$.

**Proof of Theorem 3**

Notice that if we have $\frac{\left\langle \bar{\boldsymbol{h}} - \boldsymbol{h}_{\boldsymbol{A}(k)}, \boldsymbol{h}_{i_k^*} - \boldsymbol{h}_{\boldsymbol{A}(k)} \right\rangle}{\left\| \boldsymbol{h}_{i_k^*} - \boldsymbol{h}_{\boldsymbol{A}(k)} \right\|^2} \ge 1$, then $\gamma_k^* = 1$ and in this case,

$$\left\| \boldsymbol{h}_{\boldsymbol{A}(k+1)} - \bar{\boldsymbol{h}} \right\| = \bar{\mathbb{D}}[\sigma(\boldsymbol{\theta}_{i_k^*}, \cdot), F(\cdot)] \ge \bar{\mathbb{D}}[\sigma(\boldsymbol{\theta}_{i_0^*}, \cdot), F(\cdot)].$$

On the other hand, since $0 < \left\| \bar{\boldsymbol{h}} - \boldsymbol{h}_{\boldsymbol{A}(k)} \right\|$, by the argument in proving Theorem 1, we have

$$\left\| \bar{\boldsymbol{h}} - \boldsymbol{h}_{\boldsymbol{A}(k+1)} \right\| \le \sqrt{1 - \lambda^2/D^2} \left\| \bar{\boldsymbol{h}} - \boldsymbol{h}_{\boldsymbol{A}(k)} \right\| < \left\| \bar{\boldsymbol{h}} - \boldsymbol{h}_{\boldsymbol{A}(k)} \right\|.$$

This gives that
$$\left\| \bar{\boldsymbol{h}} - \boldsymbol{h}_{\boldsymbol{A}(k+1)} \right\| < \left\| \bar{\boldsymbol{h}} - \boldsymbol{h}_{\boldsymbol{A}(k)} \right\| \le \bar{\mathbb{D}}[\sigma(\boldsymbol{\theta}_{i_0^*}, \cdot), F(\cdot)],$$

which makes contradiction.

**Proof of Theorem 4**

Using Lemma 3, we know that $\bar{\boldsymbol{h}} \in \mathrm{ri}\mathcal{M}$, which indicate that there exists some $\lambda > 0$ such that

$$\mathcal{B}(\bar{\boldsymbol{h}}, \lambda) \cap \mathrm{Aff}\mathcal{M} \subseteq \mathcal{M},$$

where $\mathcal{B}(\bar{\boldsymbol{h}}, \lambda)$ denotes the ball with radius $\lambda$ centered at $\bar{\boldsymbol{h}}$. Following the same argument of Ye et al. (2020) in proving theorem 2, we have $\left\| \bar{\boldsymbol{h}} - \boldsymbol{h}_{\boldsymbol{A}(k)} \right\|^2 = \mathcal{O}((k+1)^{-2})$. The result that $\|\boldsymbol{A}(k)\|_0 \le k+1$ is obvious as in each iteration, the number of nonzero elements in $\boldsymbol{A}$ at most increases by 1.

**Proof of Theorem 5**

Suppose that at iteration $k$, we have $h_{\boldsymbol{A}(k)}$. And the global imitation algorithm returns $h_{\boldsymbol{A}(k+1)}$ with $f_{\ell, \boldsymbol{A}(k+1)} = H_2 \circ \left[ (1 - \gamma_k) f_{\ell, \boldsymbol{A}(k)} + \sigma(\boldsymbol{\theta}_{i_k^*}, \cdot) \right] \circ H_1$. We also define $i_k'$ as the solution of local imitation. And we let $f_{\boldsymbol{A}'(k+1)} = H_2 \circ \left[ (1 - \gamma_k) f_{\ell, \boldsymbol{A}(k)} + \sigma(\boldsymbol{\theta}_{i_k'}, \cdot) \right] \circ H_1$. Define $\boldsymbol{w}_{k+1} = (k+1)(\bar{\boldsymbol{h}} - \boldsymbol{h}_{\boldsymbol{A}(k)})$, $\boldsymbol{w}_{k+1}' = (k+1)(\bar{\boldsymbol{h}} - \boldsymbol{h}_{\boldsymbol{A}'(k)})$, $\boldsymbol{W}_{k+1} = (k+1)(H \circ \bar{\boldsymbol{h}} - H \circ \boldsymbol{h}_{\boldsymbol{A}(k)})$ and $\boldsymbol{W}_{k+1}' = (k+1)(H \circ \bar{\boldsymbol{h}} - H \circ \boldsymbol{h}_{\boldsymbol{A}'(k)})$. We have

$$\begin{aligned}
\|\boldsymbol{W}_{k+1}\|^2 &\le \|\boldsymbol{W}_{k+1}'\|^2 \\
&= (k+1)^2 \left\| H \circ \bar{\boldsymbol{h}} - H \circ \boldsymbol{h}_{\boldsymbol{A}'(k)} \right\|^2 \\
&\stackrel{(1)}{\le} \kappa_1^2 (k+1)^2 \left\| \bar{\boldsymbol{h}} - \boldsymbol{h}_{\boldsymbol{A}'(k)} \right\|^2 \\
&= \kappa_1^2 \left\| \boldsymbol{w}_{k+1}' \right\|^2 \\
&\stackrel{(2)}{\le} \kappa_1^2 \left( \|\boldsymbol{w}_k\|^2 - 2\lambda \|\boldsymbol{w}_k\| + D^2 \right) \\
&\stackrel{(3)}{\le} \kappa_1^2 \left( \kappa_2^2 \|\boldsymbol{W}_k\|^2 - \kappa_2 2\lambda \|\boldsymbol{W}_k\| + D^2 \right) \\
&= \kappa_1^2 \kappa_2^2 \|\boldsymbol{W}_k\|^2 - 2\kappa_1^2 \kappa_2 \lambda \|\boldsymbol{W}_k\| + \kappa_1^2 D^2.
\end{aligned}$$

Here $D$ is the quantities defined in Lemma 5, (1) and (3) use the definition of $\kappa_1$ and $\kappa_2$ and (2) is by the argument of Ye et al. (2020) in proving Theorem 2 (notice that their argument also applies to the case that $\bar{\boldsymbol{h}}$ is in the relative interior of $\mathcal{M}$, which is proved by Lemma 3, instead

of that $\bar{h}$ is in the interior of $\mathcal{M}$). By the assumption that $D^2 \geq \kappa_1^2 \kappa_2^2 (D^2 - \lambda^2)$, the formula $\kappa_1^2 \kappa_2^2 x^2 - 2\kappa_1^2 \kappa_2 \lambda x + \kappa_1^2 D^2 = x^2$ has two real root, denoted by $z_1 \leq z_2$, where

$$z_1 = \frac{\kappa_1^2 \kappa_2 \lambda - \kappa_1 \sqrt{\kappa_1^2 \kappa_2^2 (\lambda^2 - D^2) + D^2}}{(\kappa_1^2 \kappa_2^2 - 1)}$$

$$z_2 = \frac{\kappa_1^2 \kappa_2 \lambda + \kappa_1 \sqrt{\kappa_1^2 \kappa_2^2 (\lambda^2 - D^2) + D^2}}{(\kappa_1^2 \kappa_2^2 - 1)}.$$

We define $q_1 = \kappa_1^2 \kappa_2^2$ and $q_2 = \kappa_1^2 \kappa_2$, and we have

$$\|\boldsymbol{W}_{k+1}\|^2 \leq q_1 \|\boldsymbol{W}_k\|^2 - 2q_2 \lambda \|\boldsymbol{W}_k\| + \kappa_1^2 D^2.$$

If $q_1 = 1$, then the rate holds by directly applying the argument of Ye et al. (2020) in proving Theorem 2. If $q_1 > 1$, we know that $2q_2 \lambda \geq 0$ and $\kappa_1^2 D^2 \geq 0$; $\|\boldsymbol{W}_k\| \geq 0$ for any $k$ by its definition; the formula $q_1 x^2 - 2q_2 \lambda x + \kappa_1^2 D^2 = x^2$ has two real roots $z_1 \leq z_2$; $z_2 \geq \kappa_1 D$ by the assumption; and $\|\boldsymbol{W}_{k+1}\| \leq z_2$ by the assumption. Using Lemma 7, we have, for any $k$,

$$\|\boldsymbol{W}_k\| \leq z_2,$$

which implies that

$$\left\| H \circ \bar{h} - H \circ \boldsymbol{h}_{\boldsymbol{A}(k)} \right\|^2 = \mathcal{O}((k+1)^{-2}),$$

and thus $\mathbb{D}[f_{\boldsymbol{A}(k)}, F] = \mathcal{O}(k^{-2})$. The result that $\|\boldsymbol{A}(k)\|_0 \leq k + 1$ is obvious as in each iteration, $\|\boldsymbol{A}(k)\|_0$ at most increase 1.

**Proof of Theorem 6**

When pruning the $\ell$-th layer, if this layer is pruned by local imitation, by applying Theorem 1 on the $\ell$-th layer of $f_{[\ell-1]}$, we have

$$\sqrt{\mathbb{D}[f_{[\ell]}, f_{[\ell-1]}]} = \mathcal{O}\left( \exp(-\frac{\lambda_\ell}{2} \|\boldsymbol{A}_\ell\|_0) \right),$$

for some $\lambda_\ell > 0$. Else if this layer is pruned by global imitation, we have

$$\sqrt{\mathbb{D}[f_{[\ell]}, f_{[\ell-1]}]} \leq \sqrt{\mathbb{D}[F_L \circ ... F_{\ell+1} \circ f_{\ell, \boldsymbol{A}_\ell^{\mathrm{local}}} \circ ... \circ f_{1, \boldsymbol{A}_1}, f_{[\ell-1]}]}$$

$$= \mathcal{O}\left( \exp(-\frac{\lambda_\ell}{2} \|\boldsymbol{A}_\ell^{\mathrm{local}}\|_0) \right) = \mathcal{O}\left( \exp(-\frac{\lambda_\ell}{2} \|\boldsymbol{A}_\ell\|_0) \right).$$

Using triangle inequality, we know that

$$\sqrt{\mathbb{D}[f_{[L]}, F]} \leq \sum_{\ell=1}^{L} \sqrt{\mathbb{D}[f_{[\ell]}, f_{[\ell-1]}]} = \mathcal{O}\left( \sum_{\ell=1}^{L} \exp\left( -\frac{\lambda_\ell}{2} \|\boldsymbol{A}_\ell\|_0 \right) \right),$$

with $\lambda_\ell > 0$ for all $\ell \in [L]$.

# Proof of Technical Lemmas

**Proof of Lemma 3**

The case that $n = 1$ is trivial and we consider the case that $n \geq 2$. By the definition, we know that $M$ is an non-empty and closed convex set. And thus by Lemma 2, ri$M$ is not empty. Define

$$\tilde{\boldsymbol{q}} \in \mathrm{ri}M, \ \tilde{\boldsymbol{q}} = \sum_{i=1}^{n} \alpha_i \boldsymbol{q}_i, \ \sum_{i=1}^{n} \alpha_i = 1 \text{ and } \alpha_i \geq 0 \ \forall i \in [n].$$

We define $\alpha_{\max} = \max_{i \in [n]} \alpha_i$. Notice that $\alpha_{\max} \geq 1/n$, otherwise, if $\alpha_{\max} < 1/n$, we have $\sum_{i=1}^{n} \alpha_i \leq n\alpha_{\max} < 1$, which makes contradiction. If $\alpha_{\max} = 1/n$, then $\alpha_i = 1/n$ for all $i \in [n]$, otherwise, $\sum_{i=1}^{n} \alpha_i < 1$, which makes contradiction. In the case that $\alpha_{\max} = 1/n$, we have already obtained the desired result.

Now we assume $\alpha_{\max} > \frac{1}{n}$. Define $\lambda = 1 - \frac{1}{n\alpha_{\max}} \in [0, 1)$ and $\beta_i = \frac{\alpha_{\max} - \alpha_i}{n\alpha_{\max} - 1}$. Notice this gives that

$$\sum_{i=1}^{n} \beta_i = \sum_{i=1}^{n} \frac{\alpha_{\max} - \alpha_i}{n\alpha_{\max} - 1} = \frac{n\alpha_{\max} - \sum_{i=1}^{n} \alpha_i}{n\alpha_{\max} - 1} = 1 \quad \text{and } \beta_i \geq 0 \; \forall i \in [n].$$

We define $\boldsymbol{q}' = \sum_{i=1}^{n} \beta_i \boldsymbol{q}_i$ and by the property of $\beta_i$ and the definition of $M$, we have $\boldsymbol{q}' \in M = \mathrm{cl}M$. Notice that

$$\bar{\boldsymbol{q}} = \frac{1}{n} \sum_{i=1}^{n} \boldsymbol{q}_i = (1 - \lambda) \sum_{i=1}^{n} \alpha_i \boldsymbol{q}_i + \lambda \sum_{i=1}^{n} \beta_i.$$

Using Lemma 1, we know that $\bar{\boldsymbol{q}} \in \mathrm{ri}M$.

## 5.8 Proof of Lemma 4

Notice that by choosing $s' = \bar{\boldsymbol{h}} - \lambda \frac{\bar{\boldsymbol{h}} - \boldsymbol{h}_A}{\|\bar{\boldsymbol{h}} - \boldsymbol{h}_A\|} \in \mathcal{M}$, we have

$$\max_{s \in \mathcal{M}} \langle \bar{\boldsymbol{h}} - \boldsymbol{h}_A, \bar{\boldsymbol{h}} - s \rangle \geq \langle \bar{\boldsymbol{h}} - \boldsymbol{h}_A, \bar{\boldsymbol{h}} - s' \rangle = \lambda \|\bar{\boldsymbol{h}} - \boldsymbol{h}_A\|.$$

## 5.9 Proof of Lemma 5

Notice that for any $i \in [N]$,

$$\|\boldsymbol{h}_i\| = \sqrt{\sum_{j=1}^{m} \sigma^2(\boldsymbol{\theta}_i, \boldsymbol{z}^{(j)})} \leq \sqrt{m} c_1.$$

And for any $\boldsymbol{h} \in \mathcal{M}$, we have $\boldsymbol{h} = \sum_{i=1}^{N} \beta_i \boldsymbol{h}_i$, for some $\beta_i \geq 0$ and $\sum_{i=1}^{N} \beta_i = 1$, which gives that

$$\|\boldsymbol{h}\| = \left\| \sum_{i=1}^{N} \beta_i \boldsymbol{h}_i \right\| \leq \sum_{i=1}^{n} \beta_i \|\boldsymbol{h}_i\| \leq \sqrt{m} c_1.$$

## Proof of Lemma 6

Proof of this Lemma follows standard argument in analyzing Frank Wolfe algorithm. We include it for the completeness. Notice that

$$\tilde{\boldsymbol{s}}_k^* = \arg\min_{s \in \mathcal{M}} \langle \bar{\boldsymbol{h}} - \boldsymbol{h}_{\boldsymbol{A}(k)}, s - \boldsymbol{h}_{\boldsymbol{A}(k)} \rangle = \arg\min_{s \in \mathcal{M}} \langle \bar{\boldsymbol{h}} - \boldsymbol{h}_{\boldsymbol{A}(k)}, s - \bar{\boldsymbol{h}} \rangle = -\arg\max_{s \in \mathcal{M}} \langle \bar{\boldsymbol{h}} - \boldsymbol{h}_{\boldsymbol{A}(k)}, \bar{\boldsymbol{h}} - s \rangle.$$

Using Lemma 4, we know that $\langle \bar{\boldsymbol{h}} - \boldsymbol{h}_{\boldsymbol{A}(k)}, \bar{\boldsymbol{h}} - \tilde{\boldsymbol{s}}_k^* \rangle \leq -\lambda \|\bar{\boldsymbol{h}} - \boldsymbol{h}_{\boldsymbol{A}(k)}\|$. Notice that

$$\begin{aligned}
&\langle \bar{\boldsymbol{h}} - \boldsymbol{h}_{\boldsymbol{A}(k)}, \tilde{\boldsymbol{s}}_k^* - \boldsymbol{h}_{\boldsymbol{A}(k)} \rangle \\
&= \langle \bar{\boldsymbol{h}} - \boldsymbol{h}_{\boldsymbol{A}(k)}, \bar{\boldsymbol{h}} - \boldsymbol{h}_{\boldsymbol{A}(k)} + \tilde{\boldsymbol{s}}_k^* - \bar{\boldsymbol{h}} \rangle \\
&= -\langle \bar{\boldsymbol{h}} - \boldsymbol{h}_{\boldsymbol{A}(k)}, \bar{\boldsymbol{h}} - \tilde{\boldsymbol{s}}_k^* \rangle + \|\bar{\boldsymbol{h}} - \boldsymbol{h}_{\boldsymbol{A}(k)}\|^2 \\
&\leq -2\langle \bar{\boldsymbol{h}} - \boldsymbol{h}_{\boldsymbol{A}(k)}, \bar{\boldsymbol{h}} - \tilde{\boldsymbol{s}}_k^* \rangle + \|\bar{\boldsymbol{h}} - \boldsymbol{h}_{\boldsymbol{A}(k)}\|^2 + \|\bar{\boldsymbol{h}} - \tilde{\boldsymbol{s}}_k^*\|^2 \\
&= \|(\bar{\boldsymbol{h}} - \boldsymbol{h}_{\boldsymbol{A}(k)}) - (\bar{\boldsymbol{h}} - \tilde{\boldsymbol{s}}_k^*)\|^2 = \|\boldsymbol{h}_{\boldsymbol{A}(k)} - \tilde{\boldsymbol{s}}_k^*\|^2,
\end{aligned}$$

where the last inequality uses the fact that $\langle \bar{\boldsymbol{h}} - \boldsymbol{h}_{\boldsymbol{A}(k)}, \bar{\boldsymbol{h}} - \tilde{\boldsymbol{s}}_k^* \rangle \leq 0$. This gives that $0 \leq \langle \bar{\boldsymbol{h}} - \boldsymbol{h}_{\boldsymbol{A}(k)}, \tilde{\boldsymbol{s}}_k^* - \boldsymbol{h}_{\boldsymbol{A}(k)} \rangle \leq \|\boldsymbol{h}_{\boldsymbol{A}(k)} - \tilde{\boldsymbol{s}}_k^*\|^2$. And thus we have

$$\min_{\gamma \in [0,1]} \|\boldsymbol{h}_{\boldsymbol{A}(k)} - \bar{\boldsymbol{h}}\|^2 - 2\gamma \langle \bar{\boldsymbol{h}} - \boldsymbol{h}_{\boldsymbol{A}(k)}, \tilde{\boldsymbol{s}}_k^* - \boldsymbol{h}_{\boldsymbol{A}(k)} \rangle + \gamma^2 \|\tilde{\boldsymbol{s}}_k^* - \boldsymbol{h}_{\boldsymbol{A}(k)}\|^2$$

$$= \|\boldsymbol{h}_{\boldsymbol{A}(k)} - \bar{\boldsymbol{h}}\|^2 - \frac{\langle \bar{\boldsymbol{h}} - \boldsymbol{h}_{\boldsymbol{A}(k)}, \tilde{\boldsymbol{s}}_k^* - \boldsymbol{h}_{\boldsymbol{A}(k)} \rangle^2}{\|\boldsymbol{h}_{\boldsymbol{A}(k)} - \tilde{\boldsymbol{s}}_k^*\|^2}$$

$$\leq \|\boldsymbol{h}_{\boldsymbol{A}(k)} - \bar{\boldsymbol{h}}\|^2 - \lambda^2 \frac{\|\bar{\boldsymbol{h}} - \boldsymbol{h}_{\boldsymbol{A}(k)}\|^2}{\|\boldsymbol{h}_{\boldsymbol{A}(k)} - \tilde{\boldsymbol{s}}_k^*\|^2}$$

$$\leq (1 - \lambda^2/D^2) \|\boldsymbol{h}_{\boldsymbol{A}(k)} - \bar{\boldsymbol{h}}\|^2,$$

where the last inequality is by Lemma 5.

**Proof of Lemma 7**

Define $f(x) = ax^2 - bx + c$. By assumption (1) and assumption (3), for any $z \in [z_1, z_2]$, $f(z) - z^2 \leq 0$. We proof the desired result by induction. Suppose that $x_k \in [0, z_2]$. Case 1: $x_k \in [z_1, z_2]$ and in this case,

$$x_{k+1}^2 \leq f(x_k) \leq x_k^2 \leq z_2^2.$$

Case 2: $x_k \in [0, z_1)$ and in this case

$$x_{k+1}^2 \leq f(x_1) \leq \max_{z \in [0, z_1]} f(z) \leq \max(f(0), f(z_1)) = \max(c, z_1^2).$$

This gives that $x_{k+1} \leq \max(\sqrt{c}, z_1) \leq \max(z_2, z_1) = z_2$. The desired result follows by induction.