[Reviews · NeurIPS 2020]

Review 1

Summary and Contributions: The paper proposed a greedy optimization algorithm for neural network pruning. The method has guaranteed discrepancy between pruned network and original network. The method is validated on several network structure and different tasks.

Strengths: 1. The paper proposed a new greedy optimization based prunning method with smaller error rate. 2. The paper is written clearly, and experiment is validated sufficiently.

Weaknesses: 1. More comparision and analysis on pruning compared with knowledge distill, network structure design, NAS is helpful for better understand the task. 2. Provide more details on practical efficiency of the prunning algorithm to reach a good small network

Correctness: Yes

Clarity: Yes

Relation to Prior Work: Yes. More analysis on knowledge distill, network structure design, and NAS based approach is helpful.

Reproducibility: Yes

Additional Feedback:


Review 2

Summary and Contributions: This paper proposes and analyzes greedy methods for neural network pruning. The methods are based on local or global reconstruction losses: greedily selecting neurons to include in the current layer to match the unpruned output of either the current layer or the final layer. Analysis demonstrates a better theoretical error rate than prior work, and shows competitive experimental results.

Strengths: The paper provides a simple greedy pruning algorithm (along with more complex variations for efficiency and better performance). The analysis is deeper than is typical in related papers and may provide some insight to future work. Experiments demonstrate reasonable performance on three large scale tasks.

Weaknesses: The paper is unclear in some critical areas and the overall structure of the argument takes some work to tease out (as detailed below). Further, the theoretical argument, while intriguing, seems to have little consequence in the experiments in which the loosely bound global method generally outperforms the more tightly bound local method. This raises the question of whether the long theoretical analysis was worth the effort and allocated space.

Correctness: I have not checked all proofs and derivations in the appendices. Much of what I have checked seems reasonable, however the crux of the argument lies with Theorem 1 and, as discussed below, I haven’t verified the proof. Theorem 1 claims an exponential convergence rate for global reconstruction error w.r.t. to local greedy reconstruction (while exhaustively selecting optimal neuron k at each step and computing the optimal \gamma via a convex optimization). However, each subsequent modification weakens this convergence rate from exponential to polynomial: using a fixed \gamma; selecting w.r.t. global reconstruction loss instead of local; and applying a Taylor approximation for speedup. (a) The argument that the full algorithm yields an exponential rate consists of selecting the better performing method—global or local—at each step, and then applying the local bound from Theorem 1 either directly—because the local solution was selected—or transitively—since picking the global solution means the global bound must have been too weak in that case. This broad outline of the proof structure between Theorem 1 and the convergence rate of Algorithm 1 seems to be only addressed in the supplementary section, but should be stated explicitly in the main text. (b) Given its importance, I tried to follow the proof of Theorem 1, but was repeatedly stuck on the notation. Some initial terms are undefined and just doing setup work using variables from Lemmas 1-3. However the references to \h and \bar\h confused me and I couldn’t find the right definitions to make sense of it. Perhaps this is my fault, but given the heavy lifting that this Theorem performs, it deserves a clearer proof, along with a brief intuition of the proof supplied in the main body. (c) Then there is the question of why the loosely-bounded global method would so consistently outperform the local method? Perhaps the slope is better, but the constant is worse?

Clarity: As noted above, there are things that should be clarified in the arguments for the convergence rate. In addition, there are numerous errors in grammar (e.g. verb tenses, missing conjunctions, capitalization). In addition, these items could use additional clarification: (d) Many methods apply pruning to outputs, e.g. pruning rows from FC layer or filter blocks from conv. layers. In Section 2, this method appears to prune inputs to a layer, e.g. pruning columns from an FC layer or a single filter from each filter block. These two should be equivalent in practice, but the distinction would be helpful to make explicit to avoid confusion. (Related, identifying which variables in line 53 are scalars or vectors would be clarifying.) (e) The definition and usage for U_i in line 97 is also unclear and may be incorrect. Plugging in 0.3 for a_i(k) yields [0.4, 1]. Does this make sense? (f) What are the memory requirements and algorithmic complexities for the greedy local optimization in (4), as well as subsequent alternative methods (e.g. Taylor)? (g) Line 159 mentions ancillary parameters—from the supplementary, there appears to be one additional parameter per “neuron”, denoted `b`. Is this correct? And is parameter `a`, seemingly a scale on the activation, standard in these networks? (h) As noted above, give reasoning for bound in line 164. (i) How is the method affected by batch normalization?

Relation to Prior Work: A brief summary of related work is made in the final section that broadly sets this work in related context. (j) Could the bound in Theorem 1 be adapted to other greedy methods, e.g. could the bound in Ye et al. (2020) be too loose? Specifically, how does your greedy method differ? (k) How does your Taylor method relate to the method proposed in Molchanov (2017b)? This method also uses auxiliary variables.

Reproducibility: Yes

Additional Feedback:


Review 3

Summary and Contributions: This paper proposed a greedy optimization based pruning and provided theoretical analysis to show that under some mild assumptions, sub-networks of size n with a significantly smaller O(exp(-cn)) error rate can be achieved. This work is highly related to Ye et al. (2020) and can be viewed as an extension which improves the error rates and relax the constraints in Ye et al. (2020). The proposed method is evaluated with ResNet, MobilenetV2/V3 on ImageNet and achieve better performance than Ye et al. (2020) and some other methods.

Strengths: -The contribution and novelty of this paper are clear。 -The theoretical analysis seems correct and reasonably good. -The extension of the conclusion from Ye et al. (2020) from two-layer conv nets and over-parameterized assumption to entire net with mild assumptions are meaningful and provides insights on that the pruning method is provably better than direct training with gradient descent. -Evaluation especially the ablation studies and analysis on local and global Imitation is solid and provides useful insights to better understand the method.

Weaknesses: -The discussion between the proposed method and existing ones can be improved. First, since this paper is highly related with Ye et al. (2020), it is better to specify with a little bit more details about the limitation of Ye et al. (2020) and how this paper addresses them. Meanwhile, despite the difference, the proposed local and global Imitation methods are related to some of the existing methods such as the ones that uses local or global activations to conduct network pruning. It would be nice to mention the similarity and difference among existing methods and the proposed one more explicit so the readers can better link them together. -Although the empirical experiments show better overall performance than Ye et al. (2020) and some other methods, but the margins are limited. It would be better to provide some discussions on why the theoretically significant improvements are not fully observed empirically.

Correctness: Yes, but can be improved.

Clarity: Yes, but can be improved.

Relation to Prior Work: Yes, but can be improved.

Reproducibility: No

Additional Feedback: ----After rebuttal: The authors' efforts on rebuttal are appreciated. My concerns regarding the comparison with Ye et al. (2020) are partially addressed. So I will remain my original rating and vote for acceptance.


Review 4

Summary and Contributions: This paper proposes to extend the previous GFS method as an optimization problem instead of searching. It gives a better decay rate and better accuracy of pruned models.

Strengths: The proposed method achieved much better results than previous GFS.

Weaknesses: The method of combining local and global imitation lacks theoretical analysis. Since it improves the performance, how does it affect convergence rate during pruning?

Correctness: Technically correct. However, I'm questioning about using Lipschitz continuous to prove convergence rate is tight enough analysis to explain the benefits of proposed methods.

Clarity: The paper is clearly written.

Relation to Prior Work: Previous work is well addressed. Since the proposed method is highly related to GFS, I suggested adding more ablation study to compare them in detail.

Reproducibility: Yes

Additional Feedback:

[Author Response · NeurIPS 2020]

We thanks all the reviewers very much for the constructive comments. Below please find our response. We hope you could raise your evaluation if you find that we address your concerns.

**General Response:** Reviewers ask why the practical gain of the local imitation method is not that significant as the theory. Here we give the response.

**(i)** Local imitation minimizes the surrogate local discrepancy loss while global imitation directly minimizes the original discrepancy loss. The surrogate loss is able to upper bound the original loss up to a constant but this constant still matters in practice. As a result, despite the local imitation has faster rate, both method can not consistently outperform the other one; This justifies the usefulness of comparing and choosing the better method.

**(ii)** Techniques such as [1] can be easily applied to our framework to reduce the constant gap between the two losses. However, in this paper, we aim at giving a simple algorithm which achieves SOTA both empirically and theoretically and also has future potential. We leave the further algorithm engineering for future work.

**(iii)** Our way of dealing with the Lipschitz constant is standard and tight. It is unavoidable to have such gap between theory and practice given the complexity of deep learning. But the mathematical structure we found gives stronger justification, deeper understanding of the pruning paradigm, and motivates the new algorithm, which we demonstrate useful in practice.

**Reviewer 1:** For the ImageNet experiments, the overhead of pruning phase is generally about 0.2x of that of finetuning phase. We will add more discussion on the practical computational cost of the algorithm as well as more discussion on knowledge distillation and NAS in the next version.

**Reviewer 2: (second paragraph in 'correctness')** Optimizing $\gamma$ (rather than fixing it) is a core component to achieve faster convergence rate. However, using global reconstruction loss, does not necessarily weaken the exponential rate, if we still optimize $\gamma$. The proof framework of Theorem 4 can be extended to this case (optimizing $\gamma$ and minimize global reconstruction loss) and gives exponential decay rate. We will add more discussion on that. Besides, Taylor approximation is not used in local imitation as each iteration is already fast (line 102-107) and thus does not effect the exponential rate. **(a, h)** We will move this statement to main text. **(b)** Lemma 1-3 are self contained in terms of notation. We guess that you find cl$M$ and ri$M$ undefined? Their definition is in the beginning of Appendix (the def of $h$ and $\bar{h}$ are also there). The main intuition is that local imitation actually enjoys good geometric property (line 551), which makes imitating the internal layer's output very efficient (see key inequality between line 555 and 556). **(c)** Please see general response. **(d)** Yes, it is equivalent in practice. The boldsymbol denotes vector. We will improve the clarity. **(e)** Sorry for the typo. $U_i = [-a_i(k)/(1 - a_i(k)), 1]$. **(f)** The time and space complexity is small. For local imitation, we only add one extra parameter for each neuron and this parameter will be merged into the scale of neuron after the pruning finish. Following in line 102-107 and sec 5.1 in appendix, executing the pruning algorithm only requires one forward pass and the selection can be done with simple matrix multiplication using O(batch size × num of channels) space complexity. The Taylor approximation is only applied to global imitation, which only adds 2 extra parameters for each neuron. The main time complexity for selecting one neuron is calculating the gradient of the ancillary variable, which is also small. See sec 5.4 for more details. **(g, i)** Yes, $b$ is the ancillary parameters. But $a$ in the appendix is identical to the $a$ used in the main text (e.g. between line 61 and 62). And $a$ is not the scale on the activation but $w_1$ is (see line 53). In practice, we regard the weight, activation and batchnorm together as a neuron. **(j)** The bound in Theorem 1 can not be adapted to Ye at al. (2020) and their bound is tight. One key difference is that we optimizing $\gamma$ during selection, which significantly enlarge the search space of each greedy step and thus improves the rate. **(k)** They are very different. Our method is based on forward selection (starting from empty network and greedily add neurons). Taylor approximation is only a speed-up technique for us. In comparison, Molchanov (2017b) eliminates neurons from full network by looking at neuron importance measured by Taylor expansion.

**Reviewer 3: (1)** Our theoretical improvement over Ye et al. (2020) is discussed in introduction and Table 1. In terms of algorithm, intuitively, our method weights each selected neurons differently instead of simply average them like Ye et al. (2020), which brings the improvement. We will give more detailed discussion on difference between Ye et al. (2020) and other existing papers. **(2)** There are two reasons. Firstly, the final comparison is made after finetuning and thus part of the improvement on pruning phase might be smaller after finetuning. For the second point, please see general response.

**Reviewer 4:** Please see general response (iii) on the 'Lipschitz continuous' comment. We will add more ablation studies similar to that in sec 2.3 and sec 3.1 to compare with GFS.

**Reference:** [1] Zhuang, Zhuangwei, et al. "Discrimination-aware channel pruning for deep neural networks." Advances in Neural Information Processing Systems. 2018.

[Meta-Review · NeurIPS 2020]

The paper proposes a method for neuron pruning with sound theoretical guarantees and interesting insights. There are concerns that the important proofs could be made more clear - the authors should try to address this in a final version.